# Hydrological impacts of climate change on small ungauged catchments-results from a GCM-RCM-hydrologic model chain

Aynalem T. Tsegaw[1], Marie Pontoppidan[2], Erle Kristvik[1], Knut Alfredsen[1], Tone M. Muthanna[1]

[1] Department of Civil and Environmental Engineering, Norwegian University of Science and Technology (NTNU), S.P. Andersensvei 5, N-7491, Trondheim, Norway.

[2] NORCE Norwegian Research Centre, Bjerknes Centre for Climate Research, Bergen, Norway.

Correspondance to: Aynalem T. Tsegaw (aynalem.t.tasachew@ntnu.no)

**Abstract.** Climate change is one of the greatest threats to the World's environment. In Norway, the change will strongly affect the pattern, frequency and magnitudes of stream flows. However, it is highly challenging to quantify to what extent it will affect flow patterns and floods from small ungauged rural catchments due to unavailability or inadequacy of hydro-meteorological data for

the calibration of hydrological models and tailoring methods to a small-scale level. To provide meaningful climate impact studies at small catchments, it is therefore beneficial to use high spatial and temporal resolution climate projections as input to a high-resolution hydrological model. Here we use such a model chain to assess the impacts of climate change on flow patterns and frequency of floods in small ungauged rural catchments in western Norway using a new high-resolution

regional climate projection, with improved performance with regards to the precipitation distribution, and the regionalized hydrological model (Distance Distribution Dynamics) between the reference period (1981-2011) and a future period (2071-2100). The FDCs of all study catchments show there will be more wetter periods in the future than the reference period. The results also show that in the future period, the mean annual flow increases by 16.5% to 33.3%, and

there will be an increase in the mean autumn, mean winter and mean spring flows ranging from



4.3% to 256.3%. The mean summer flow decreases by 7.2% to 35.2%. The mean annual maximum

floods increase by 28.9% to 38.3%, and floods of 2 to 200 years return periods increase by 16.1%

to 42.7%. The findings of this study could be of practical use to regional decision-makers if

considered alongside other previous and future findings.

# 1  Introduction

Climate change is one of the greatest threats to human existence, economic activity, ecosystems

and civil infrastructures (Kim and Choi, 2012). Climate change risks for natural and human

systems are higher for global warming of 1.5 $^{0}$C than at present, but lower than at 2 $^{0}$C, and these

risks depend on the magnitude and rate of warming, geographic location, levels of development,

vulnerability, and on the choices and implementation of adaptation and mitigation options (IPCC,

**2018**). The trend of changes in different parts of Europe may vary considerably because of changes

in large-scale atmospheric circulation or local orographic circulation (Eisenreich et al., 2005,

Hattermann et al., 2007)..

Changes in temperature and precipitation and the shift in winter precipitation from snow to rain

will be crucial in studying impacts of climate change on hydrology of a catchment. These changes

influence the hydrological regime of a stream, and the most series and widespread potential impact

of the changes is flooding (Baltas, 2007, Richardson, 2002, Thornes, 2001). In Norway, the

average annual temperature and precipitation are expected to increase by 3.8 ℃ to 6.2 ℃ and 7%

to 27% respectively by the end of the century using RCP8.5 emission scenario (Hanssen-Bauer et

al., 2015). The largest increases in precipitation are mostly expected during the autumn and winter



months and will in turn impact the magnitude and in some cases the seasonality of peak runoff and floods. A climate impact study at Sogn and Fjordane county of Norway showed that flood peaks

shifted from summer to autumn for the future scenario (Chernet et al., 2014), and Donnelly et al. (2017) studied climate change impacts on European hydrology and found that in the Norwegian region, climate change will strongly affect the hydrological cycle in the future period. Also, outside Norway, authors have reported that the frequency and magnitude of flows are being affected by the changes in climatic conditions (Alfieri et al., 2015; Madsen et al., 2014; Mallakpour &

Villarini, 2015; Rojas et al., 2013). Climate change adverse results upon streamflow regimes worldwide (Pumo et al., 2016), calls for attention of the impact study on a local scale.

Projected increase in the frequency and intensity of heavy localized precipitation events, based on climate models, contributes to increasing in precipitation-generated local flooding, and an increase

in local sudden flooding is causing significant danger and loss of life and property (Borga et al., 2011, Kundzewicz et al., 2014). Local sudden floods (flash floods) usually occur in small catchments (e.g., catchments less than $100 - 1000$ km$^2$). This type of flood event is usually short in duration, but it is usually connected with severe damage (Menzel et al., 2006). Studies show that the probability and magnitude of hazardous heavy precipitation events have been increasing

in several European regions e.g., (Golz et al., 2016). Heavy localized precipitation could be caused by low pressure system (e.g., western Norway (Azad and Sorteberg, 2017)) or because of prevailing convective precipitation at hilly or mountainous areas.

A quantitative analysis of the impacts of climate change on flooding conditions requires

simulations of climatological-hydrological system. The models on which the simulations are based



should give an adequate representation of the system dynamics relevant for different types of flow (e.g. floods) generation (Menzel et al., 2006). The climate and hydrological models are the two models involved in climatological-hydrological system. Climate change affects the basic components of hydrologic cycles, and the application of hydrological models provide the means

to conceptualize and investigate the relationship between climate (e.g. precipitation and temperature) and water resources (e.g. low flows and floods) of a region to assess the likely effects of climate change and propose appropriate adaptation strategies (Baltas, 2007). The regional impacts of climate change (e.g. on local flooding) come out with the necessity of orienting adaptation measures to local climatic, geographic, economic and social conditions (Hattermann,

2009, Krysanova et al., 2008). Because catchment storage (e.g. effective lake, soil and groundwater) has a significant role in altering the timing between the precipitation and runoff, the relationship between projected changes in precipitation and corresponding runoff cannot be compared directly. Therefore, hydrological modelling based on a local climate scenario is required to assess the impact that changes in precipitation and temperature will have on processes leading

to different types of flows (e.g. floods) (Lawrence et al., 2012). This is generally performed by following a sequence of steps from global and regional climate modelling, through data tailoring (downscaling and bias-adjustment) and hydrological modelling (Olsson et al., 2016).

Climate impact assessment on hydrology of small ungauged catchments using continuous

hydrological modelling is challenging because of unavailability or inadequacy of hydro-meteorological data for calibration of hydrological models, short response time of the catchments, difficulty in describing local hydrological processes and coarse resolution of climate models. The challenge in coarse spatial resolution of climate models is due to poor representation of



precipitation which is inadequate for assessment of impacts on smaller catchments (Quintero et

al., 2018). For example, Pontoppidan et al. (2017) showed that during a flooding event in western

Norway, the regional model simulated observed rainfall considerably better with a grid spacing of

3 km compared to a grid spacing of 9 km due to the complex terrain in the area. Therefore,

to provide a meaningful climate impact results at small catchments, it is necessary to use high

spatial and temporal resolutions of projected climate data that can be used as forcing in

high resolution hydrological models (Lespinas et al., 2014; López-Moreno et al., 2013; Reynolds

et al., 2015; Tofiq & Guven, 2014). Current efforts of coordinated regional downscaling in Europe

(EURO-CORDEX e.g. (Jacob et al., 2014; Kotlarski et al., 2014)) are performed on a 0.11° grid,

however a new high-resolution regional downscaling with improved representation of local

precipitation distribution for southern Norway is available (Pontoppidan et al., 2018), but has yet

to be included in a full hydrological model chain.

To solve the challenge related to lack of availability of a properly calibrated high-resolution

hydrological model at ungauged small rural catchments in Norway, a predictive tool has been

developed and tested. Tsegaw et al. (2019a) calibrated and validated Distance Distribution

Dynamics (DDD) hydrological model at forty-one gauged small rural catchments in Norway with

hourly temporal resolution. For predicting flow at ungauged catchments, the DDD model

parameters have been regionalized using three methods of regionalization (multiple regression,

physical similarity and combined method) and the methods have been tested on seven independent

catchments. The finding shows that the combined method performs the best of all the methods in

predicating flow. Even if the DDD model predicts flow at ungauged catchments satisfactory (0.5

≤ Kling-Gupta Efficiency < 0.75), the model underestimates most of the observed flood peaks. To



improve the prediction of observed floods, a dynamic river network method has been introduced and implemented in DDD (Tsegaw et al., 2019b). This improved setup has been used in this study where the general objective is to assess the hydrological impacts of climate change on small
ungauged catchments using a novel model chain consisting of a high resolution, bias corrected dynamical downscaling and the improved DDD model. We specifically focus on:

i.  Assessing the impacts of climate change on the changes of flow patterns at ungauged small rural catchments around Bergen, Norway.

ii.  Assessing impacts of climate change on the pattern and frequency of floods in
ungauged small rural catchments around Bergen, Norway.

The knowledge gained is critical for decision makers so that flood risk management strategies can be planned accordingly and in a timing manner.

## 2 Data and methods

**2.1 Study area**

Six ungauged small rural catchments, located in western Norway around Bergen, are used in this study. We selected the catchments using http://nevina.nve.no/ and https://www.norgeskart.no/. The definition of small rural catchments is based on the report of Fleig and Wilson (2013) with an upper area limit of 50km$^2$. The catchments are selected for the impact study because there are
critical infrastructures (e.g. culverts, bridges and buildings) at the outlet of the catchments which could be affected by the climate change in the future period. We selected three catchments with bare mountain dominated (>50%) and three catchments with forest dominated (>50%) to include diverse land uses in the study. The locations and observed river networks of the selected


catchments are depicted in Fig. 1. The catchment descriptors (CDs) and outlet coordinates of each

study catchment are presented in Table 1.

## 2.2 Climate, topography and land use data

The precipitation and temperature data used to drive the hydrological model are obtained from a

simulation performed by the Weather Research and Forecasting model (WRF) version 3.8.1

(Skamarock et al., 2008). The model is non-hydrostatic and widely used for weather forecasting

and research purposes. The simulation has a spatial grid resolution of 4 km x 4 km and the

precipitation and temperature are available every 3 hours.  However, regional models, as WRF,

inherit biases from the boundary conditions used to drive the model. These biases may lead to

misrepresentation of important features in the models, e.g. the known bias of the North Atlantic

storm track (Zappa et al., 2013) leads individual storms into central Europe instead of a more

northern path along the Norwegian coast as observations suggest. Therefore, the global climate

model NorESM1-M used as forcing data at the boundaries in WRF was corrected for such biases

before the regional downscaling. This led to a more realistic representation of the North Atlantic

storm track and the precipitation distribution in southern Norway (Pontoppidan et al., 2018).


The DDD model parameters, which do not need calibration, are derived from an analysis of hydro-

meteorological, topographical and land use data of a catchment using GIS. The source of the

topography and land use data is the Norwegian Mapping Authority (www.statkart.no). The 10m x

10 m DEM, the river network and the 1: 50 000 scale land use data have been retrieved and used

in the study. The DEM has been re-conditioned to the naturally occurring river network using  the

Arc-hydro tool to create a hydrologically correct terrain model that can  improve the accuracy of


watershed modeling (Li, 2014). The re-conditioned DEM is further used to determine the distance

distributions of hill slopes and river networks as needed by DDD.

## 2.3 DDD hydrological model


The Distance Distribution Dynamics (DDD) hydrological model is developed by Skaugen and

Onof (2014) and currently runs operationally with daily and three-hourly time steps at the

Norwegian flood forecasting service. It has two main modules: the subsurface and the dynamics

of runoff.

### 2.3.1 The Subsurface


The volume capacity of the subsurface water reservoir, M (mm), is shared between a saturated

zone with volume, S (mm), and an unsaturated zone with volume, D (mm). The volume of the

saturated zone and the unsaturated zone are inversely related i.e. the higher the unsaturated zone

volume, the lower the saturated zone (Skaugen and Mengistu, 2016, Skaugen and Onof, 2014).

The actual water volume present in the unsaturated zone is described as Z (mm). The subsurface

state variables are updated after evaluating whether the current soil moisture, Z(t), together with

the input snowmelt and rain, G(t), represent an excess of water over the field capacity, R, which is

fixed at 30 % of D(t) i.e. R = 0.3 (Skaugen and Onof, 2014). If G(t) + Z(t) > R*D(t), then the

excess water X(t) is added to S(t).

Excess water $\quad\quad\quad\quad X(t) = Max \left\{ \frac{G(t)+Z(t)}{D(t)} - R, 0 \right\} D(t) \quad\quad \left[ \frac{mm}{3 hours} \right] \quad\quad (1)$

Groundwater $\quad\quad\quad\quad \frac{dS}{dt} = X(t) - Q(t) \quad\quad\quad\quad\quad\quad \left[ \frac{mm}{3 hours} \right] \quad\quad (2)$

Soil water content $\quad\quad\quad \frac{dZ}{dt} = G(t) - X(t) - Ea(t) \quad\quad\quad \left[ \frac{mm}{3 hours} \right] \quad\quad (3)$



| | | | |
|---|---|---|---|
| Soil water zone | $\frac{dD}{dt} = -\frac{dS}{dt}$ | $\left[\frac{mm}{3hours}\right]$ | (4) |
| Potential evapotranspiration | $Ep = Cea * T$ | $\left[\frac{mm}{3hours}\right]$ | (5) |
| Actual evapotranspiration | $Ea = Ep * \frac{S+Z}{M}$ | $\left[\frac{mm}{3hours}\right]$ | (6) |


### 2.3.2 Runoff dynamics

The dynamics of runoff in DDD has been derived from the catchment topography using a GIS combined with runoff recession analysis. In DDD, the distribution of distances between points in

the catchment and their nearest river reach (distance distributions of a hillslope) is the basis for describing the flow dynamics of the hillslope. The distribution of distances between points in the river network and the outlet forms the basis for describing the flow dynamics of the river network. The hillslope and river flow dynamics of DDD is hence described by unit hydrographs (UHs) derived from distance distributions from a GIS and celerity derived from recession analysis

(Skaugen and Mengistu, 2016, Skaugen and Onof, 2014). When the distance distributions are associated with flow celerity of the hillslope and rivers, we obtain the distributions of travel times which constitutes the time area concentration curve (Maidment, 1993). The derivative of the time area concentration curve gives an instantaneous unit hydrograph (UH) (Bras, 1990), which is basically a set of weights distributing the input (precipitation and snowmelt) in time to the outlet.


Previous studies in more than 120 catchments in Norway showed that the exponential distribution describes the hillslope distance (Euclidean distance from the nearest river reach) distribution well, and the normal distribution describes well the distances between points in the river network and outlet of a catchment (Skaugen and Onof, 2014). Figure 2 shows the structure of the DDD model.





The model is written in the R programming language. All GIS work is done with ArcGIS 10.3

(ESRI, 2014), and the recession analysis is done using a R script (R Core Team, 2017) .

### 2.3.3 Dynamic river network method in DDD

Dynamic expansions and contractions of stream networks play an important role for hydrologic

processes since they connect different parts of the catchment to the outlet (Nhim, 2012).  Dynamic

river networks and hence dynamic overland unit hydrographs are introduced and implemented in

the DDD model to improve the simulation of floods (Tsegaw et al., 2019b). The mean of the

distribution of distances from a point in the catchment to the nearest river reach ($D_m$) becomes

dynamic in the dynamic river network method. We therefore need to estimate the dynamic $D_m$

from the relation between upstream critical supporting area ($A_c$) i.e. the area needed to initiate and

maintain streams and $D_m$ using GIS as shown in Eq.(7). Coefficients a and b are estimated for each

study catchments and are presented in Table 2. The calibration parameter of the dynamic river

network routine in DDD is critical flux ($F_c$) and is estimated by regional regression.

$$D_m = aA_c{}^{\boldsymbol{b}} \tag{7}$$



### 2.3.4 Model parameters


The DDD model parameters are divided into three main groups. The first group are those estimated from observed hydro-meteorological data (for gauged catchments) or through regionalization for ungauged catchments (appendix 1), the second group are those estimated by model calibration (for gauged catchments) against observed discharge or by regionalization methods (for ungauged catchments) (appendix 2), and the third group are those estimated from digitized geographic maps


using a GIS (appendix 3). The snow routine in DDD has two parameters estimated from the spatial distribution of observed precipitation data (Skaugen and Weltzien, 2016). The shape parameter (a0) and the decorrelation length (d) of the gamma distribution of snow and snow water equivalent (SWE) are estimated from a previous calibration for 84 catchments in Norway (Skaugen et al.,


2015). Since our study focuses on ungauged catchments, we cannot conduct calibration, and we therefore derived the model parameters needing calibration through combined method of regionalization using 41 gauged small rural catchments in Norway as a base (Tsegaw et al., 2019a).

### 2.3.5 Regionalizing the parameters of DDD model


To estimate the regionalized parameters for this study (3 hourly time step), we have used the combined method of regionalization which has been recommended for estimating regionalized DDD model parameters with hourly resolution (Tsegaw et al., 2019a). In the combined method of regionalization, we have estimated the recession parameters and critical flux using multiple regression between model parameters and CDs, and the other parameters (all in appendix 2) using


the physical similarity method with pooled donor catchments. The parameters of the model needing regionalization are shown in appendix 1 and 2 (the bottom 5 parameters in appendix 1 and all in appendix 2).





The CDs of the study catchments, used for multiple regression, are presented in Table 1. The

multiple regression equations used in this study are taken from the above-mentioned references

and presented below.

$$Gscale = \exp(-5.12 - 0.12Le + 0.22\ln(S_q) + 0.3\log(M_e)) \quad (8)$$

$$Gshape = 0.82 + 0.0005M_p - 0.009S_q \quad (9)$$

$$GshI = 2.047Gshape - 0.658 \quad (10)$$

$$GscI = 0.49Gscale - 0.0014 \quad (11)$$

$$f_c = 160.7 - 1.4B \quad (12)$$

The parameters of DDD model needing calibration are estimated using a pooling-group type of

physical similarity method of regionalization. Kay et al. (2006) defined physical similarity using

Euclidean distance in a space of CDs as shown in Eq.13.

$$dist_{a,b} = \sqrt{\sum_{j=1}^{J}\left(\frac{X_{a,j} - X_{b,j}}{\sigma_{x,j}}\right)^2} \quad (13)$$

Where $j$ indicates one of a total of 12 CDs (all in Table 1 except outlet locations), $X_{a,j}$ is the value

of that CD at the $a^{th}$ ungauged catchment (the six catchments in Table 1), $X_{b,j}$ is the value of the

CD at b[th] catchment which is a member of the 41 calibrated catchments studied in Tsegaw et al.

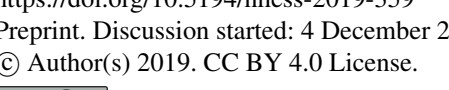



(2019a) , and $\sigma_{x,j}$ is the standard deviation of the CD across all the 41 catchments. Seven closest neighbor catchments (minimum distance) are selected to create a pooling group. After identifying the pooling group members, we have computed the model parameter at the ungauged catchment $a$ ( $\alpha_a{}^{PG}$) as a weighted average of the parameters of the 7 members. Kay et al. (2007) stated that it

is more appropriate to write the expression for the model parameter as a weighted average of the estimated parameter values, $\alpha_m$, for all 41catchments (N) as shown in Eq. (14).

$$\alpha_a{}^{PG} = \frac{\sum_{m=1}^{N} h_{am}\alpha_m}{\sum_{m=1}^{N} h_{am}} \qquad (14)$$

Catchments not in the pooling group are given a weight $h_{am}$ equal to zero, but those in the pooling are assigned weights to reflect their importance which is based on the distance measure $dist_{a,b}$ as

defined in Eq. (13). The weights of the pooling group members are estimated by Eq. (15).

$$h_{am} = 1 - S_{am} \qquad (15)$$

where

$$S_{am} = {dist_{a,b}}\Big/{dist_{a,max}} \qquad (16)$$

where $dist_{a,max}$ is set to be 10% larger than the maximum distance of a pooling group member

from the ungauged catchment $a$.

## 2.4 Impact study

We have extracted the precipitation and temperature data from the 4 X 4 km and 3 hourly resolution climate model. The climate data are forced into the DDD model to simulate the runoff,





actual evapotranspiration and snow water equivalent (SWE) both for the reference and future

periods. We have used 30 hydrological years (1st of September to 31st of August) for both periods

of the impact study. We have analyzed changes of the following climate impact indicators:

     i)      The mean annual changes of precipitation, temperature, flow, snow water equivalent

         (SWE) and actual evapotranspiration.

ii)      The mean annual and mean seasonal changes of flow

     iii)      The annual and seasonal flow duration curves (FDCs)

     iv)      The timing of annual winter/spring and fall stream flow

     v)      The mean annual and seasonal maximum flows

     vi)      Floods with return periods of 2 to 200 years

Changes are computed by Eq. (17) using the magnitudes of hydro-climatic variable for the

reference and future periods.

$$Change\ in\ x\ (\%) = \left( \frac{Future\ value\ of\ x - Reference\ value\ of\ x}{Reference\ value\ of\ x} \right) * 100 \qquad (17)$$

where $x$ is any hydro-climatic variable.

**2.4.1 Changes of hydro-climatic elements**

The 3 hourly precipitation and temperature data, extracted from the climate model, are analyzed

using an R-script to quantify the changes in the mean annual values for the reference and future

periods. The 3-hourly precipitation data are aggregated yearly to estimate the annual precipitation

value and then averaged over the 30 years to get the mean annual value. The 3-hourly temperature

data are averaged for the whole 30 years to estimate the mean annual temperature. The simulated





3-hourly flow is averaged for the whole 30 years to get the mean annual flow data. Seasonal mean flow data are also estimated for the reference and future periods i.e. winter, spring, summer and autumn for assessing changes in the seasonal mean flow. The annual maximum SWE is selected from each hydrological year and averaged for reference and future periods to get the mean annual

maximum SWE for the two periods. The annual actual evapotranspiration is estimated by aggregating the actual evapotranspiration from the 3-hour simulation results and then averaged over 30 years to get the mean annual actual evapotranspiration.

### 2.4.2 Changes in flow duration curves

A flow duration curve is a cumulative curve that shows the percent of time a specified flow is equaled or exceeded during a given period, and it shows the flow characteristic of a stream throughout a range of flow, without regard to the sequence of occurrence (Searcy, 1959). We have analyzed changes in the stream flow variability over a water year between the reference and future periods. The changes of floods (between 0% and 10% exceedance), medium flows (between 10%

and 70% exceedance) and low flows (between 70% and 100% exceedance) are analyzed in this study. The formula to calculate the probability of exceedance is given by Eq. (18).

$$p = 100 * {}^{K}\!/\!{}_{(n + 1)} \qquad (18)$$

$p$ = the probability that a given flow will be equaled or exceeded (% of time)

$K$ = the ranked position on the listing (dimensionless)

$n$ = the number of events for period of record, and it is dimensionless



### 2.4.3 Changes in timing of annual winter/spring and fall stream flow

The annual timing of river flows is a good indicator of climate-related changes. Changes in timing of annual winter/spring (WS) and fall stream flow is analyzed using center of volume date

(Hodgkins et al., 2003). The center of volume date is the date by which half of the total volume of water for a given period flows by a river section. The center of volume date is expected to be more robust indicator of the timing of the bulk of high flows in a season than the peak flows, as the peak flow may happen before or after the bulk of seasonal flows (Hodgkins et al., 2003). From the 3-hour flow data (simulated for the reference and future periods), we have calculated the mean 3-

hour flow for the 30 years in both periods. Using the mean 3-hour flow, we have computed seasonal center of volume dates for the winter/spring (1 January to 31 May) and fall (1 October to 31 December).

### 2.4.4 Changes in the maximum flows and flood frequency

The annual and seasonal maximum flows (floods) are selected from the 30 years of reference and future periods for the analysis. The changes in the mean and median of the annual and seasonal maximum flows are analyzed.

The number of 3-hour floods (frequency) above a certain threshold helps us to have a general overview on the impacts of climate change on the flood risk in small catchments. Accordingly, we

have analyzed the changes in the number of 3-hour floods between the reference and future periods with a flow higher than the minimum of the 30 years annual maximum flow for the reference period.





To assess the magnitude of a flood with a given probability, flood frequency methods must be

applied. Flood frequency analysis is important for flood hazard mapping, for which a flood of a

certain return period (e.g. 200 years in Norway) is used for the flood zone mapping (Groen et al.,

2012). To analyses changes in the magnitudes of a flood with a given return period (e.g. 200-year

flood), flood frequency analysis is applied to the annual maximum series for the reference (1981

– 2011) and future periods (1970 – 2100). The percentage change in the flood magnitude is then

computed as the difference between the two curves divided by the flood magnitude for the

reference period. We have used a Gumbel distribution (Bhagat, 2017, Shaw, 1983) to model the

annual maximum series in this study. We have selected the Gumbel distribution because it has

been widely applied including the studies of climate change impacts on floods in Europe (Dankers

and Feyen, 2008, Veijalainen et al., 2010).


## 3  Results

### 3.1 Regionalized DDD model parameters

The results of the parameters values from the regionalization for the six study catchments are

presented in Table 3. The parameters and possible ranges of values are presented in appendix 4.

### 3.2 Changes in hydro-climatic elements

The simulation results of the hydrological model are further analyzed to quantify the changes in

the hydro-climatic elements. The mean annual precipitation, the mean annual temperature, the

mean annual evapotranspiration, the mean annual flow, the mean autumn flow, and the mean

winter flow increase for all the study catchments in the future period compared to the reference

period. The mean spring flow increases in the five catchments and decrease in one study

catchment. The mean summer flow decreases for the five catchments. The mean annual maximum SWE decreases for all the study catchments. In the future period, the mean annual precipitation increases by 20% to 23.9 %. The mean annual temperature rises in 3 - 3.3 degree Celsius. The mean annual flow increases from 16.5% to 33.3%. The decrease in the mean summer flow ranges

from 7.2% to 35.2% and the increase is 3.6% in only one of the study catchments. The mean winter flow increases by an average of 126.9% (ranging from 41.3% to 256.3%). The mean spring flow increases by 4.3% to 99.7% for the five catchments and there will be a decrease by 1.4% in one catchment. The mean autumn flow increases by an average of 37% (ranging from 20.6% to 43.9%). The results of changes of the mean annual temperature, precipitation, maximum SWE and actual

evapotranspiration are presented in Table 4. Table 5 presents changes in the mean annual and seasonal flows for the catchments. Mean 3 hourly flow of the study catchments are shown in Fig. 3 for the reference and future periods.

### 3.3 Changes in flow duration curves

The results of the study show that changes in the flow duration curves (FDCs) values are positive for all the flow conditions. The FDC values of the future period increase for all flow conditions (low, medium and high flows) for all the study catchments. For all catchments, the top 5% of flows in the future period are higher than the reference period by 7.6% to 61.5%. The median flow (flows which are exceeded by 50% of the time) increases by 23.7% to 139.6% (the highest value is for

catchment 1 and the lowest value is for catchment 4) in the future period. Figure 4 shows the FDCs for both periods.





### 3.4 Changes in timing of annual winter/spring (WS) and fall stream flow

For all the study catchments, the mean WS center of volume dates occur earlier in the future period

(16 - 68 days) than the reference period. The fall CV date occurs later for all the study catchments

in the future period and a shift of 1 – 16 days is expected. Table 6 presents the mean WS CV dates

and mean fall CV dates for all the study catchments.

### 3.5 Changes in the maximum flows and flood frequency

### 3.5.1 Changes in the annual and seasonal maximum flows

The annual and seasonal maximum flows increase in the future period compared to the reference

period. The mean annual maximum flows increase from 28.9% to 38.3% across all the study

catchments. The mean seasonal maximum flows also show an increase in all seasons (1.1 % to

118%) and all catchments except for spring season of catchment 2 (reduction of 28.9%) as shown

in Table 7. The median of the annual and seasonal maximum flows increases for all catchments

except for spring season of catchment 2 as shown in Fig.5. Table 7 presents the results of changes

in the mean annual and seasonal maximum flows in future period compared to the reference period.

Figure 5 shows the distributions of the 30 years annual and seasonal maximum flows both for the

reference and future periods.


The number of 3-hours with floods exceeding the minimum annual maximum flood in the 30 years

of the reference period increases in the future period significantly (Table 8). This result shows that



flooding will occur more often in the future period. In the future period, the yearly average number

of such floods increase between 61.7% to 133% across all study catchments.

**3.5.2 Changes in flood frequencies**

The flood frequency analysis using Gumbel's Extreme Value Distribution shows that floods of 2,

5, 10, 20, 25, 50, 100 and 200 years return periods increase in the future period (2070 – 2100)

compared to the reference period (1981 – 2011) for all catchments. The increase ranges from

16.1% to 42.7%. Table 9 shows the changes of flood frequencies for the selected return periods

for all the study catchments.

## 4 Discussion

### 4.1 Regionalized DDD model parameters

When we estimate the DDD model parameters needing calibration using the pooling group method

of regionalization for the ungauged study catchments, many of the most similar gauged catchments

(from the 41 database) are found to be in the western Norway (west climate region) and close to

the ungauged study catchments which shows that the regionalization method used in this study is

plausible.

### 4.2 Hydrological impacts of climate change

#### 4.2.1 Changes of hydro-climatic elements

Generally, the findings of the increase in precipitation and temperature for the study catchments

are in the range of increments predicted by the Norwegian Center for Climate Services (NCCS)

under the report Climate in Norway 2100 (Hanssen-Bauer et al., 2015) ; however the results from




some catchments are above or below the prediction interval of the report since the comparison is

between catchments specific results with the regional values of the report. The NCCS report is

based upon ten climate models with RCP8.5 and RCP4.5 using daily temporal resolution, and we

have compared our findings with the RCP8.5 results of the report. The report shows that there will

be an increase of precipitation by 2.5% to 21% for the Hordaland county of Norway (where our

study catchments are located) between 1971-2000 and 2071 – 2100 and there will be an increase

of temperature by 3.1 to 4.9 degree Celsius for ensembles of RCP 8.5. The results of the climate

model in our study are generally in agreement with the aforementioned report i.e. changes of 20%

- 23.9% and 3-3.3 degree Celsius for precipitation and temperature respectively; however, the

results from the climate model used in this study predicts precipitation changes to the higher end

of the climate service report and the temperature changes towards the lower end of the climate

service report. In the future period, all the study catchments show an increase in the mean annual

flow when compared to the reference period. The maximum increase is 33.3%, and the minimum

increase is 16.5%. Alcamo et al. (2007) found that mean annual river flow projected to increase in

northern Europe (e.g. Norway) by approximately 9% to 22% up to the 2070 which aligns with our

findings i.e. the increment could increase by 16.5% up to 33.3 % to 2100. The increase in mean

annual flow (mean annual water volume) in the future period is a result of a substantial increase in

projection of the mean annual precipitation with a moderate increase in mean temperature i.e. the

mean annual precipitation increases by 20% to 23.9% while the mean annual temperature increases

by $3^{\circ}c$ to $3.3^{\circ}c$ (Table 4). The increase in the mean annual temperature results in an increase of

water loss by evapotranspiration. However, the mean annual increase in precipitation exceeds the

mean annual increase in the actual evapotranspiration (43% to 131.5%) and these conditions

contributed to increase of mean annual flow in general. The Climate in Norway 2100 report shows
that the mean annual flow for western Norway (where the study catchments are located) could increase from -1% to 17% in 2100 and our result shows that the increase is slightly higher than the increase in the report for four of the study catchments. This could be related to the capability of

the climate model used in this study to reflect the local representation of precipitation and temperature, the differences in the temporal resolution used by this study and the report and the averaging issue in estimating the regional value by the report.

Unlike the changes in the mean annual flow, changes in the temporal distribution of flows (e.g.

seasonal) can be important because changes are rarely identical throughout the year (Olsson et al., 2016). The mean winter and autumn flows increase for all study catchments. The main causes of increases are projected increase in the precipitation and temperature during the autumn and winter seasons. The increase in mean winter flow contributes to much of the increase in the mean annual flow for all catchments (Table 5 and Fig.3). The main cause of increase in the mean winter flow

is increased winter temperatures. Increased winter temperatures result in a higher proportion of winter precipitation to fall as rain which then results in a higher proportion of winter flow. The mean spring flows show an increase for the five catchments and a decrease for one catchment while the mean summer flows show a decrease for the five catchments and an increase for one of the catchments.


Similar results are found in other hydrological assessments of the Bergen region. Previous studies of the water resources in Bergen under climate change also project higher temperatures and increased annual precipitation in the Bergen region for the 2071-2100 future period under the




RCP8.5 emissions scenario (Kristvik et al., 2018, Kristvik and Riisnes, 2015). Kristvik et al. (2018)

based their assessment on statistical downscaling of an ensemble of RCPs and GCMs, followed

by simulations of the hydrological response in term of inflow to surface water reservoirs. Due to

higher temperatures and more rainfall precipitation, strong increases in winter flow was found,

while a decrease was projected in spring/summer months due to less snowmelt (Kristvik et al.

2018).


The climate in Norway 2100 report for western Norway shows that the mean winter and autumn

flow increase by 15% to 42% and by 5% to 36% respectively in 2100. The findings of this study

show that the increase in mean winter flow is higher than the maximum prediction in the report

for four catchments and to the higher end of the prediction in the report for the remaining two

catchments. Similar results have been obtained for mean autumn flows except that three

catchments have higher value than the maximum prediction value in the report. The report predicts

an increase of the mean spring flow by -9% to 17% and a decrease of mean summer flow by 13%

to 28% in 2100. The findings of this study show that the increase in the mean spring flow is within

the prediction interval of the report for three catchments and higher than the maximum prediction

value of the report for the rest three catchments. The results of this study show a higher decrease

than the maximum prediction in the report for the three catchments and lower decrease than the

minimum prediction in the report for two catchments. Only one catchment shows a decrease with

in the prediction interval of the report. Wong et al. (2011) studied the differences in hydrological

drought characteristics in summer season of Norway between the periods 1961-1990 and 2071-

2100 using HBV hydrological model with daily temporal resolution and found that substantial

increases in hydrological drought duration and drought affected areas are expected in Norway

which aligns with our findings. Ministry of the Environment of Norway (2009) also pointed out that the summer flow in Norway is projected to be reduced and supports the findings of our study.

Climate change affects snow pack and the amount of water stored in the snow pack (SWE). Increased winter temperature will generally lead to a reduction in snow storage and hence the mean maximum SWE will also reduce in the future. The results of this study show that there will be a reduction in the mean maximum SWE at all the catchments in the future period. The reduction ranges from 47.5% to 77.8%. The largest reduction is found to be at the catchment with the highest

mean elevation value (catchment 1). Snow accumulation and its characteristics are the results of air temperature, precipitation, wind and the amount of moisture in the atmosphere. Therefore, changes in these and other climatic properties can affect snow pack and hence maximum SWE. In our study, there is an increase in precipitation and temperatures for all study catchments in the future period, and the increase resulted in the reduction of mean annual maximum SWE at all the

study catchments.

### 4.2.2 Changes in Flow duration curves (FDCs)

The results of this study show that climate change affects the FDCs of the study catchments. The future FDC is higher than the FDC of the reference period at all catchments for all probability of

exceedances (Fig.4). The FDCs of all the study catchment show that the low flows increase in the future, and there will be more wetter periods in the future than in the reference period.





### 4.2.3 Changes in WSCV and fall CV dates

The mean winter/spring center of volume date (WSCV) will be earlier, and the mean fall CV date

will be later for all the study catchments. The change in WSCV dates is related to the amount and

timing of spring snowmelt and warmer winter temperature. The earlier mean WSCV date in the

future period is the result of increased precipitation falling during a warmer winter, reduced snow

storage, early snow melt and warmer spring temperature. The late occurrence of fall CV dates is

related to the higher precipitation and temperature projected in fall in the future period. The warmer

temperature in the future period makes the major proportion of future precipitation to fall as rain

which in turn increases the total flow volume in fall which makes the fall CV dates to occur later.

### 4.3 Changes in the maximum flows and flood frequency

### 4.3.1 Annual and seasonal maximum flows

In the future period (2070 – 2100), the results of this study show that there will be an increase in

the mean and median of the annual and seasonal maximum flows (Table 7 and 8 and Fig.5) at all

the study catchments except for the spring season of catchment 2.  Many (15 – 23 of the 30 annual

maximum floods) of the maximum annual flows happen during the autumn period (1st September

to 30th of November) and therefore much of the contribution for the increment of the mean and

median annual maximum flows comes from the autumn (Fig.5). The second higher contribution

for the increment of the mean and median annual maximum flows is winter season (Fig.5). In the

future period, the winter maximum flows increase in magnitude and frequencies as a substantial

amount of precipitation falls as rain in a warmer climate. The mean summer maximum flows show

the least increment in the future period (1.1% to 20.7%), but the summer season contributes to the



increment of the mean and median of the annual maximum flows (the third higher contributor to

the annual maximum flows). The mean spring maximum flows show the highest increment in

percentage (25.4% to 118% for five of the study catchments and 28.9% reduction for one of the

study catchments) compared to the other seasons, but the contribution of the spring season for the

mean and median increment of the annual maximum flows is the least of all the seasons. The

finding that mean annual maximum flows (floods) increase by 28.9% to 38.3% in our study is

supported by Lawrence and Hisdal (2011). Lawrence and Hisdal (2011) have done ensemble

modelling based on locally adjusted precipitation and temperature data from 13 regional climate

scenarios to assess likely changes in hydrological floods between a reference period (1960 – 1990)

and two future periods (2021-2050) and (2071 - 2100), for the 115 catchments distributed

throughout in Norway. Their results showed that western regions of Norway are associated with

the largest percentage increases in the magnitude of the mean annual floods (> 20%). Lawrence

and Hisdal (2011) also pointed out that increase in autumn and winter rainfall throughout Norway

will increase the magnitude of peak flows during these seasons, and at areas already dominated by

autumn and winter floods, the projected increases in floods magnitude will be large. Since our

study catchments are at western Norway which is dominated by autumn floods and our finding

(Fig.5) confirms their finding in that the maximum increases in floods magnitude are expected to

happen in autumn and winter seasons (Table 7 and Fig.5).

The yearly average number of 3-hours flows, which are greater than the minimum of the annual

maximum high flows in the 30 years of the reference period increases. The yearly average number

of such floods increase between 61.7% and 133% across all study catchments as presented in Table

8. The results show that there will be a greater number of 3-hours floods in the future period than



the reference period, and more flood risks are expected at the infrastructures constructed downstream of small ungauged rural catchments in west coast Norway near Bergen city. European

Environmental Agency(2004b), in: Alcamo et al. (2007) found that the risk of floods increases in northern Europe (e.g. Norway) which supports our finding of increase in the risk of floods. Center for International Climate Research (https://cicero.oslo.no) predicts that western Norway will experience more heavy rain and flooding in the future and our findings confirms their predictions.

**4.3.2 Changes in flood frequency analysis**

The study results from the six ungauged small catchments show that there will be an increase in flood frequencies with a return periods of 2, 5, 10, 20, 25, 50, 100, 200 years in the future period. The changes of all return periods for all catchments are in between 16.1% and 42.7%. The maximum and the minimum changes happen for a return period of 200 years. For all return periods,

the mean changes are between 31 % and 32% while the median changes are between 30% and 34%. The 2, 5, 10 years changes are greater than 20% for all catchments and 20, 25, 50, 100 and 200-years changes are greater than 20% for five of the six study catchments.

Beldring et al. (2006) studied the percentage change in the mean annual flood and the 50-year

flood in four catchments in Norway between 1961-1990 and 2070-2100 in which one of the catchments is in western Norway (Viksvatn in Gaular) and found that moderate to large increases are expected. Their result is supported by our finding i.e. the 50-year flood on six small catchments in west Norway will increase by 18.2% to 40 % between 1981 -2011 and 2070-2100. In our study results, the 200-year flood changes are 16.1%, 34.7%, 41.3%, 42.7%, 31.1% and 22.7% for





catchments 1, 2, 3, 4, 5, 6 respectively. Lawrence and Hisdal (2011) have found that the projected increase of the 200-year flood exceed 40% for some of the catchments in western Norway between the 1961-1990 reference period and the 2071- 2100 future period which is in agreement with our findings. Lawrence (2016) used ensembles of regional climate projections from EURO-CORDEX together with HBV model to assess possible effects of climate change on floods on 115 catchments in Norway for two future periods (20131-2016 and 2071-2100). The assessment result shows that the minimum increase in the 200 years flood for catchments less than 100km$^2$ at Hordaland county (where the study catchments are located) is 20% which is generally in agreement with our findings.

## 4.4 Limitations

A possible uncertainty related to hydrological modelling in this study is that we have used the regionalization model developed for 1 hour (Tsegaw et al., 2019a) to estimate parameters for the 3-hour simulation used in this study. DDD model parameters like degree hour factor for evapotranspiration (Cea) and degree hour factors for snow melt (Cx) can be sensitive to the temporal resolution. As presented in Table 4, the mean annual actual evaporation value has smaller result than what is expected for Norway which is the result of low value of Cea. However, the same uncertainty is present both in the reference and future periods. A second possible limitation is that DDD model parameters are assumed to be constant under changing climatic conditions, and the same parameter sets are used for the reference and future period simulations. However, studies show that using the same parameter sets for the reference and future periods under climate impact studies can have significant impact on the simulation results. Merz et al. (2011) found that the impact on simulated flow of assuming time invariant hydrological model parameters can be very significant. Thirdly, the modelled changes in the hydro-climatic elements and flood frequency are

derived from a single GCM-RCM model chain, however this simulation has the benefit of a high spatial resolution for a better representation of small-scale features and additionally a novel bias

correction method has been applied. Other combinations of GCMs and RCMs predict varieties of future climate change signals which could potentially result in different hydro-climatic and flood predictions for the same study catchments. Ensemble simulations are needed to fully understand and address the uncertainties of future changes in the hydroclimatic elements, however a single model study, like we have used in this study, increases our knowledge and understanding.

Therefore, the results of this study alone should not be taken as a conclusive of what will be seen in the future but could be of practical use to regional decision-makers if considered alongside other previous and future findings.

## 5  Conclusion

In this study we use a bias corrected dynamical downscaling product as input for the DDD model to investigate the impact of climate change on small ungauged catchments. The results show that there will be an increase in the mean annual flow in the future period. The increase in the mean annual flow is due to the increase in the mean autumn, winter and spring flows in the future period (2070-2100) compared to the reference period (1981 - 2011). In the future period, the mean

summer flows from the study catchments decrease. Future flow duration curves are higher than the flow duration curves of the reference period for all study catchments for all probability of exceedances. The median flow (flows which are exceeded by 50% of the time) increases by 23.7% to 139.6%. The FDCs of all the study catchment show that the low flows increase in the future, and there will be more wetter periods in the future than in the reference period.


There will be an increase in the mean annual floods and flood frequencies of 2, 5, 10, 20, 25, 50, 100 and 200 years in the future period. The mean annual maximum floods increase by 28.9% to 38.3%. This study gives clear indication that the projected increase in flood frequencies are high (e.g. 200-year flood > 40%) in small catchments around Berge area of western Norway, and such

catchments are vulnerable to an increased risk in the future climate. The high-resolution regional climate model with a novel bias correction method improves the knowledge and understanding of climate change impacts on hydrology of small catchments in western Norway. However, it is important to conduct further researches which can address the limitations of this study before conducting flood risk assessment and planning flood risk management strategies as a national

strategy for climate change adaptation.

These simulations are based on high resolution regional climate model projection with a novel bias correction method and address limitations in previous impact studies where such projections have not yet been available and enabling in-depth analysis of the impacts of climate change on rapid

hydrological processes. An ensemble of GCM-RCM runs building on the results of this paper is suggested as a venue for further work in order to account for uncertainties in future emissions and climate projections and thus provide more reliable recommendations for infrastructure design and adaptation.




## Acknowledgments

The authors would like to acknowledge Norwegian Climate Service Centre for providing information on how to access and process the 3 X 3 km spatial and 3-hours temporal resolution gridded precipitation and temperature climate data for western Norway. The authors also would like to acknowledge Thomas Skaugen of the Norwegian Water Resources and Energy Directorate (NVE) for providing the source code of Distance Distribution Dynamics hydrological model. Finally, the authors gratefully acknowledge the financial support by Bingo and ExtndBingo project (EU Horizon 2020, grant agreement 641739), and the Research Council of Norway (RCN) through R3 (grant 255397) and through the Centre for Research-based Innovation "Klima 2050" (see www.klima2050.no).





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




# Appendixes

**Appendix 1.** List of DDD model parameters estimated from observed precipitation data and those estimated from regionalization (multiple regression) for the study catchments.

| Parameters | Description of the parameter | Method of estimation | Unit |
|---|---|---|---|
| d | Parameter for spatial distribution of SWE, decorrelation length | From spatial distribution of observed precipitation | Positive real number |
| a0 | Parameter for spatial distribution of SWE, shape parameter | From spatial distribution of observed precipitation | Positive real number |
| MAD | Long term mean annual discharge | Specific runoff map of Norway | $m^3\,sec^{-1}$ |
| Gshape | Shape parameter of $\lambda$ | Regression | Positive real number |
| Gscale | Scale parameter of $\lambda$ | Regression | Positive real number |
| GshInt | Shape parameter of $\Lambda$ | Regression | Positive real number |
| GscInt | Scale parameter of $\Lambda$ | Regression | Positive real number |
| Fc | Critical flux | Regression | $m^3$/hour |




**Appendix 2.** List of DDD rainfall-runoff model parameters estimated from pooling group of

physical similarity method of regionalizations.

| Parameters | Description of the parameter | Method of estimation | Unit |
|---|---|---|---|
| Pro | Liquid water in snow | Regionalization (poolig group) | fraction |
| Cx | Degree hour factor for snow melt | Regionalization (poolig group) | mm $^{\circ}$C$^{-1}$ hour$^{-1}$ |
| CFR | Degree hour factor for refreezing | Regionalization (poolig group) | mm $^{\circ}$C$^{-1}$ hour$^{-1}$ |
| Cea | Degree hour factor for evapotranspiration | Regionalization (poolig group) | mm $^{\circ}$C$^{-1}$ hour$^{-1}$ |
| rv | Celerity for river flow | Regionalization (poolig group) | m/s |






**Appendix 3.** List of DDD rainfall-runoff model parameters estimated from geographical data using GIS.

| Symbol of parameters | Description of the Parameter |
|---|---|
| area | Catchment area |
| maxLbog | Maximum distance of marsh land portion of hillslope |
| midLbog | Mean distance of marsh land portion of hillslope |
| bogfrac | Areal fraction of marsh land from the total land uses |
| zsoil | Areal fraction of DD for soils (what area with distance zero to the river) |
| zbog | Areal fraction of distance distribution for marsh land (what area with distance zero to the river) |
| midFl | Mean distance (from distance distribution) for river network |
| stdFL | Standard deviation of distance (from distance distribution) for river network |
| maxFL | Maximum distance (from distance distribution) for river network |
| maxDl | Maximum distance (from distance distribution) of non-marsh land (soils) of hill slope |
| midDL | Mean distance (from distance distribution) of non-marsh land (soils) of hill slope |
| midGl | Mean distance (from distance distribution) for Glacial |
| stdGl | Standard deviation of distance (from distance distribution) for Glacial |
| maxGl | Maximum distance (from distance distribution) for Glacial |
| Hypsographic curve | 11 values describing the quantiles 0, 10, 20, 30, 40, 50, 60,70,80,90,100 |





**Appendix 4.** Possible ranges of regionalized DDD model parameters

| Model parameters needing regionalization | Method of regionalization | Possible ranges of values |
|---|---|---|
| Gshape | Multiple regression | Positive real number |
| Gscale | Multiple regression | Positive real number |
| GshInt | Multiple regression | Positive real number |
| GscInt | Multiple regression | Positive real number |
| fc | Multiple regression | Positive real number |
| Pro | Pooling group type of physical similarity | 0.03 - 0.1 |
| Cx | Pooling group type of physical similarity | 0.05 - 1.0 |
| CFR | Pooling group type of physical similarity | 0.001 - 0.01 |
| Cea | Pooling group type of physical similarity | 0.01 - 0.1 |
| rv | Pooling group type of physical similarity | 0.5 - 1.5 |



**Figure Captions**

**Figure 1.** Locations of study catchments in Norway

**Figure 2.** Structure of the Distance Distributions Dynamics model adapted from Skaugen and Onof (2014). Left panel: the storage model and right panel: hydrographs of hillslope and river

**Figure 3.** Yearly mean 3 hourly hydrographs of the study catchments for the reference and future periods

**Figure 4:** Flow duration curves (FDCs) of the 3-hourly flow for the six study catchments both for the reference and future periods

**Figure 5.** Distributions of the annual and seasonal maximum flow values of the 30 years period
**Figures**



**Figure 1.** Locations of study catchments in Norway.

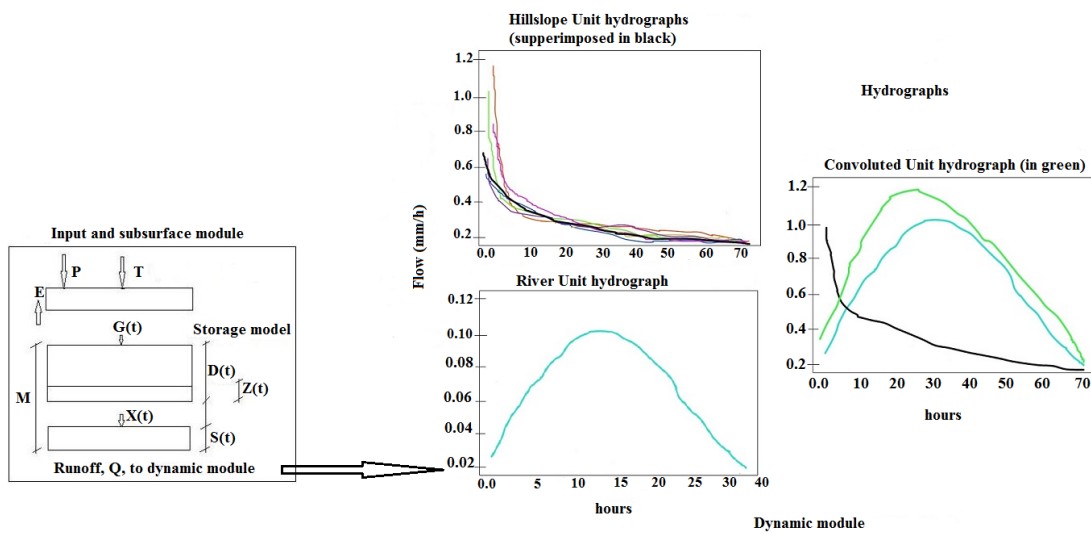

**Figure 2.** Structure of the Distance Distributions Dynamics model adapted from Skaugen and Onof (2014). Left panel: the storage model and right panel: hydrographs of hillslope and river.

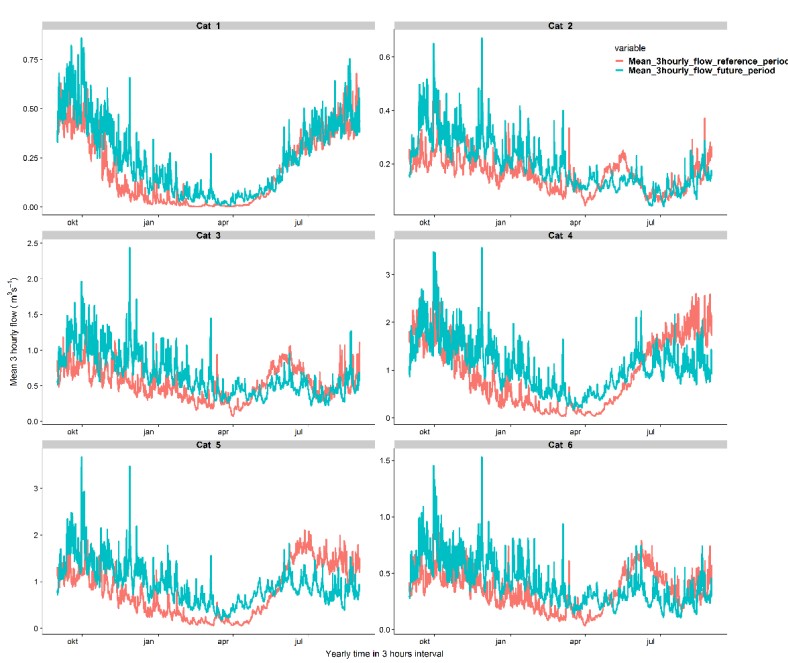

**Figure 3.** Yearly mean 3 hourly hydrographs of the study catchments for the reference and future periods



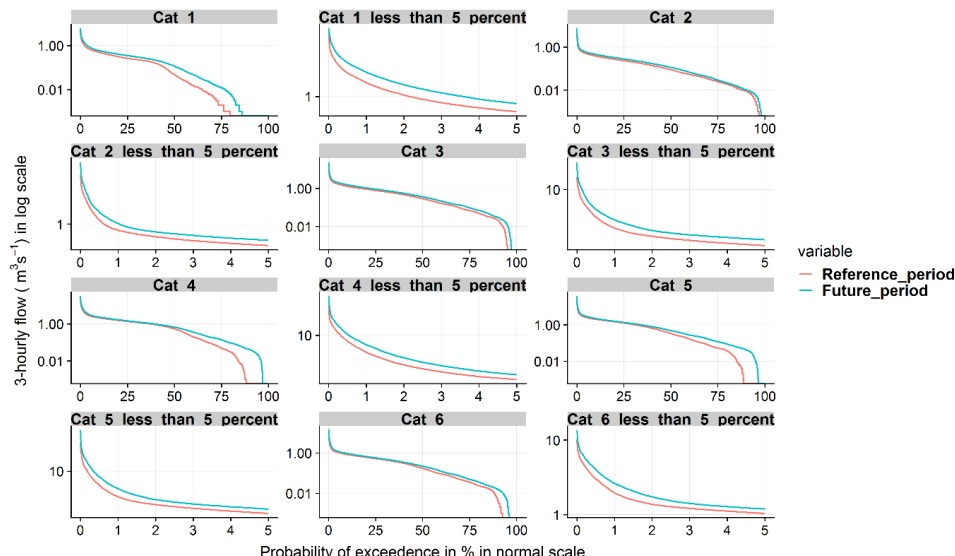

**Figure 4:** Flow duration curves (FDCs) of the 3-hourly flow for the six study catchments both for the reference and future periods

Natural Hazards
and Earth System
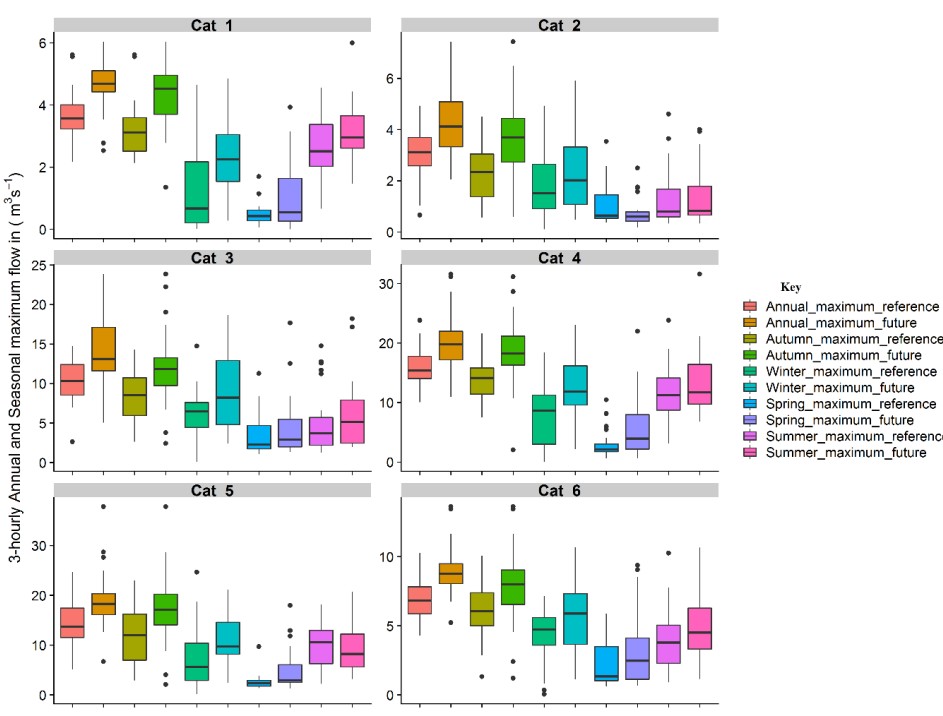

**Figure 5.** Distributions of the annual and seasonal maximum flow values of the 30 years
perio




**Tables**

**Table 1:** Catchment descriptors of the study catchments

| Catchments Descriptors | Unit | Symbol | Catchments | | | | | |
| --- | --- | --- | --- | --- | --- | --- | --- | --- |
| | | | Cat_1 | Cat_2 | Cat_3 | Cat_4 | Cat_5 | Cat_6 |
| Mean of distance distributions of soils in the catchment to the nearest river reach | $m$ | $D_m$ | 103.0 | 169.1 | 204.3 | 137.0 | 174.9 | 171.7 |
| Mean of distance distributions of marsh land in the catchment to the nearest river reach | $m$ | $D_{mr}$ | 0.0 | 261.0 | 220.7 | 109.9 | 107.2 | 154.3 |
| Mean of distance distribution of points in the river to the outlet | $m$ | $D_r$ | 1513.2 | 960.5 | 2671.2 | 3061.1 | 3402.8 | 1733.3 |
| Catchment area | $km^2$ | A | 1.5 | 2.3 | 7.3 | 7.9 | 8.2 | 3.8 |
| Effective lake | % | $L_e$ | 0.2 | 0.0 | 0.0 | 0.7 | 0.0 | 0.0 |
| Forest | % | F | 18.5 | 65.3 | 75.8 | 22.5 | 69.7 | 25.4 |
| Bare mountain | % | B | 79.6 | 27.6 | 14.8 | 66.0 | 18.9 | 65.3 |
| Urban | % | U | 0.0 | 0.1 | 0.0 | 0.0 | 0.0 | 0.0 |
| Mean elevation | $m$ | $M_e$ | 684.6 | 322.1 | 314.7 | 461.5 | 402.1 | 466.7 |
| Mean anual precipitation | $mm$ | $M_p$ | 3268.0 | 2243.0 | 2500.0 | 2781.0 | 2543.0 | 2644.0 |
| Speciifc discharge | $l\ s^{-1} km^{-2}$ | $S_q$ | 141.0 | 115.7 | 91.8 | 125.6 | 134.2 | 110.7 |
| Mean river slope | $m\ km^{-1}$ | $R_s$ | 162.6 | 266.2 | 88.4 | 106.4 | 118.6 | 154.9 |
| **Outlet location** | | | | | | | | |
| ETRS_1989_UTM_Zone_33N coordinate system (m) | | Longtiude | -9376.0 | -14513.6 | -15886.7 | -22440.2 | -14280.8 | -25871.8 |
| | | Latitude | 6777231.6 | 6712810.0 | 6758694.5 | 6725236.5 | 6719015.4 | 6732970.8 |




**Table 2:** Coefficients of the power relation between $D_m$ and $A_c$ and the coefficients of determination (R-squared).

| Catchment_ID | a | b | R-squared |
|---|---|---|---|
| Cat_1 | 1.42 | 0.41 | 0.97 |
| Cat_2 | 0.87 | 0.45 | 0.99 |
| Cat_3 | 0.87 | 0.46 | 1 |
| Cat_4 | 1.2 | 0.44 | 0.99 |
| Cat_5 | 0.99 | 0.45 | 1 |
| Cat_6 | 0.87 | 0.46 | 1 |



**Table 3:** DDD model parameters of the study catchments estimated from regionalization

| Model parameters needing regionalization | Catchments | | | | | |
|---|---|---|---|---|---|---|
| | Cat_1 | Cat_2 | Cat_3 | cat_4 | Cat_5 | cat_6 |
| Gshape | 2.317 | 1.827 | 1.977 | 2.087 | 1.961 | 2.032 |
| Gscale | 0.041 | 0.036 | 0.034 | 0.033 | 0.038 | 0.037 |
| GshInt | 4.085 | 3.083 | 3.39 | 3.615 | 3.356 | 3.502 |
| GscInt | 0.018 | 0.016 | 0.015 | 0.015 | 0.017 | 0.017 |
| fc | 49.3 | 122.1 | 140.00 | 68.30 | 134.2 | 69.00 |
| Pro | 0.1 | 0.087 | 0.082 | 0.100 | 0.095 | 0.096 |
| Cx | 0.155 | 0.129 | 0.108 | 0.137 | 0.159 | 0.147 |
| CFR | 0.004 | 0.006 | 0.007 | 0.004 | 0.003 | 0.004 |
| Cea | 0.033 | 0.025 | 0.016 | 0.032 | 0.028 | 0.031 |
| rv | 1.22 | 1.240 | 1.17 | 1.200 | 1.260 | 1.190 |



**Table 4:** Changes of mean annual temperature and precipitation, mean annual maximum snow

water equivalent (SWE) and mean annual evapotranspiration for all the study catchments

| Hydro-meteorological indicator | Unit | Change in indicator |
|---|---|---|
| *Cat_1* | | |
| Mean annual precipitation | mm | 22.2 % |
| Mean annual temprature | °c | 3.3 °c |
| Mean annual maximum SWE | mm | -77.8 % |
| Mean annual evapotranspiration | mm | 62.8 % |
| *Cat_2* | | |
| Mean annual precipitation | mm | 23.9 % |
| Mean annual temprature | °c | 3.1 °c |
| Mean annual maximum SWE | mm | -47.5 % |
| Mean annual  evapotranspiration | mm | 66.5 % |
| *Cat_3* | | |
| Mean annual precipitation | mm | 23.64 % |
| Mean annual temprature | °c | 3.2 °c |
| Mean annual maximum SWE | mm | -49.81 % |
| Mean annual  evapotranspiration | mm | 43 % |
| *Cat_4* | | |
| Mean annual precipitation | mm | 20.4 % |
| Mean annual temprature | °c | 3.2 °c |
| Mean annual maximum SWE | mm | -56.05 % |
| Mean annual evapotranspiration | mm | 131.5 % |
| *Cat_5* | | |
| Mean annual precipitation | mm | 22.1 % |
| Mean annual temprature | °c | 3.2 °c |
| Mean annual maximum SWE | mm | -48.6 % |
| Mean annual evapotranspiration | mm | 80.5 % |
| *Cat_6* | | |
| Mean annual precipitation | mm | 20.0 % |
| Mean annual temprature | °c | 3.0 °c |
| Mean annual maximum SWE | mm | -63.0 % |
| Mean annual evapotranspiration | mm | 91.8 % |




**Table 5:** Changes in percentage of mean annual flow and seasonal flows of the study catchments. The unit of the flows is m³/s.

| Hydrologic indicator (flow) | Change in indicator (%) | Hydrologic indicator (flow) | Change in indicator (%) |
|---|---|---|---|
| *Cat_1* | | *Cat_4* | |
| Mean annual flow | 33.3 | Mean annual flow | 16.5 |
| Mean winter flow | 256.3 | Mean winter flow | 167.7 |
| Mean spring flow | 48.9 | Mean spring flow | 99.7 |
| Mean summer flow | 3.6 | Mean summer flow | -32.7 |
| Mean Autumn flow | 43.9 | Mean Autumn flow | 20.6 |
| *Cat_2* | | *Cat_5* | |
| Mean annual flow | 21.9 | Mean annual flow | 18.9 |
| Mean winter flow | 41.3 | Mean winter flow | 146.7 |
| Mean spring flow | -1.4 | Mean spring flow | 76.4 |
| Mean summer flow | -7.2 | Mean summer flow | -41.0 |
| Mean Autumn flow | 37.8 | Mean Autumn flow | 43.3 |
| *Cat_3* | | *Cat_6* | |
| Mean annual flow | 21.9 | Mean annual flow | 17.0 |
| Mean winter flow | 68.3 | Mean winter flow | 81.1 |
| Mean spring flow | 4.3 | Mean spring flow | 10.0 |
| Mean summer flow | -21.2 | Mean summer flow | -35.2 |
| Mean Autumn flow | 41.1 | Mean Autumn flow | 35.1 |





**Table 6:** Winter/spring and fall center of volume dates for the six study attachments

| Annual timing | Center volume (CV) date | Center volume (CV) date | Is CV date early or late? |
|---|---|---|---|
| *Cat_1* | | | |
| Winter/Spring | 13 May | 5 March | early |
| Fall | 21 October | 31 October | late |
| *Cat_2* | | | |
| Winter/Spring | 18 March | 2 March | early |
| Fall | 11 November | 12 November | late |
| *Cat_3* | | | |
| Winter/Spring | 27 March | 3 March | early |
| Fall | 8 November | 11 November | late |
| *Cat_4* | | | |
| Winter/Spring | 24 April | 10 march | early |
| Fall | 29 October | 8 November | late |
| *Cat_5* | | | |
| Winter/Spring | 26 April | 13 March | early |
| Fall | 3 November | 19 November | late |
| *Cat_6* | | | |
| Winter/Spring | 11 April | 3 March | early |
| Fall | 8 November | 11 November | late |



**Table 7:** Changes in percentage of the mean annual and seasonal maximum flows in the future

period compared to the reference period.

| Annual and Seasonal maximum flows | Change in indicator (%) | Annual and Seasonal maximum flows | Change in indicator (%) |
|---|---|---|---|
| *Cat_1* | | *Cat_4* | |
| Mean autumn maximum flow | 37.7 | Mean autumn maximum flow | 33.1 |
| Mean winter maximum flow | 82.4 | Mean winter maximum flow | 59.8 |
| Mean spring maximum flow | 118.0 | Mean spring maximum flow | 105.5 |
| Mean summer maximum flow | 16.7 | Mean summer maximum flow | 17.7 |
| Mean annual maximum flow | 28.0 | Mean annual maximum flow | 28.9 |
| *Cat_2* | | *Cat_5* | |
| Mean autumn maximum flow | 60.0 | Mean autumn maximum flow | 48.2 |
| Mean winter maximum flow | 32.2 | Mean winter maximum flow | 48.6 |
| Mean spring maximum flow | -28.9 | Mean spring maximum flow | 86.4 |
| Mean summer maximum flow | 7.2 | Mean summer maximum flow | 1.1 |
| Mean annual maximum flow | 38.3 | Mean annual maximum flow | 31.4 |
| *Cat_3* | | *Cat_6* | |
| Mean autumn maximum flow | 43.2 | Mean autumn maximum flow | 27.5 |
| Mean winter maximum flow | 45.7 | Mean winter maximum flow | 28.9 |
| Mean spring maximum flow | 25.4 | Mean spring maximum flow | 41.3 |
| Mean summer maximum flow | 20.7 | Mean summer maximum flow | 26.8 |
| Mean annual maximum flow | 36.9 | Mean annual maximum flow | 28.9 |


**Table 8:** Changes in the number of 3-hour floods which are greater than the minimum annual maximum flood in the reference period for all the study catchments.

| Catchment ID | Mean annual number of 3-hours floods greater than the minimum annual maximum flood in the reference period | | Changes in number (%) |
|---|---|---|---|
| | Reference period (1981-2011) | Future period (2070-2100) | |
| *Cat_1* | 9.1 | 21.2 | 133.0 |
| *Cat_2* | 58 | 99.3 | 71.2 |
| *Cat_3* | 38 | 64.4 | 69.5 |
| *Cat_4* | 9 | 15.4 | 71.1 |
| *Cat_5* | 22.2 | 35.9 | 61.7 |
| *Cat_6* | 7 | 13.3 | 90 |




**Table 9:** Changes of flood frequencies with return periods of 2, 5, 10, 20, 25, 50, 100 and 200 years between the future and reference periods using Gumbel's Extreme Value Distribution for all study catchments.

| T(years) | Change (%) | | | | | |
|---|---|---|---|---|---|---|
| | Cat_1 | Cat_2 | Cat_3 | Cat_4 | Cat_5 | Cat_6 |
| 2 | 28.9 | 36.7 | 35.7 | 28.0 | 31.4 | 29.5 |
| 5 | 24.1 | 35.9 | 37.9 | 33.3 | 31.3 | 26.9 |
| 10 | 21.8 | 35.5 | 38.9 | 35.9 | 31.2 | 25.7 |
| 20 | 20.0 | 35.3 | 39.7 | 38.0 | 31.2 | 24.8 |
| 25 | 19.5 | 35.2 | 39.9 | 38.6 | 31.2 | 24.5 |
| 50 | 18.2 | 35.0 | 40.5 | 40.2 | 31.1 | 23.8 |
| 100 | 17.0 | 34.9 | 40.9 | 41.5 | 31.1 | 23.2 |
| 200 | 16.1 | 34.7 | 41.3 | 42.7 | 31.1 | 22.7 |