# Peer review of "Hydrological impacts of climate change on small ungauged catchmentsresults from a GCM-RCM-hydrologic model chain"

_Natural Hazards and Earth System Sciences, 2019_

## Referee Comment (RC1) · Anonymous Referee #1 · 12 Jan 2020

In the manuscript the regionalized DDD model with dynamic river network was used to study climate change impacts on hydrology by 2070-2100 in ungauged catchments in the Bergen area in western Norway. Six ungauged small rural catchments are modeled with a single high-resolution downscaled climate scenario.

The manuscript is well structured and well written and bring new knowledge on estimation of climate change impacts on small ungauged basins with sub-daily time steps. However, to improve the manuscript, the results, which use input from one climate model, should be given more context in regard to other climate scenarios and some key uncertainties should be better presented.

[Figure]

Detailed comments

Abstract: The abstract should clearly state the climate scenario (especially rcp) for which the results are based since this affects the likelihood of the proposed changes (ie. these are apparently with the rcp8.5 and are therefore likely to be the upper end of the proposed changes). The uncertainties due to use of only one climate scenario should be shortly acknowledged also in the abstract and the percentage changes of the results should be provided with less accuracy (ie. not well 256.3 % but with e.g. 260 %).

Data and methods

More background information form the Bergen area floods and the mechanism (snow or rain or both) could be provided. In discussions the results could be reflected with this.

Section 2.2

Section 2.2 includes information of the climate change input data used. However the climate change input data should be described in more detail -The climate scenario used should be more clearly stated (Global climate model, regional climate model, rcp) (NORESM-M, WSF, but the rcp is not mentioned here, apparently rcp8.5) - the corrections made to the data should be provided with more details. Is only the GCM corrected and for which variables? Is there any bias correction on the RCM data? How well do the temperatures and precipitation compare to observations? -the use of only one scenario just be better justified (since the common approach these days in to use several scenarios to enable uncertainties to be included). Why this particular model and why only one?

Discussion

Since only one climate scenario is used, the influence of this decision on the results should be discussed. • Table 4 states the temperature and precipitation changes

with the used climate scenario (the GCM-RCM and rcp of the scenario should be added to the table header) and on page 2 the range for rcp8.5 is stated. However these should be compared more clearly in the discussion or elsewhere (how does the chosen scenario compare to others, is it e.g. wetter than average). Also the range of temperature and precipitation with other rcps than just 8.5 should be provided for context. The results are currently only been compared to the rcp8.5 results of e.g. NCCS report, also some comparison with the rcp4.5 could be provided for more context. • How does the use of only one scenario influence the results and what are the likely results with other scenarios (e.g. are these likely to be the top end of changes in floods which can be used as worst case scenario). Currently the results, which are stated with high accuracy, can provide false sense of certainty while this major uncertainty is not well established. (The emissions used in RCP8.5 pathway are nowadays considered by some scientists as rather unlikely due to the ongoing mitigation efforts and the sinking prices of renewables. Therefore there has been arguments against the use of this scenario as a "business as usual" scenario).

The results showed large increase in flood risk due to climate change. Other studies are referred to but the main differences between these studies explaining the differences in the results (the inclusion of different types of catchments with more snow dominated flooding and the use of several different climate scenarios) should be analyzed.

The changes in max SWE (table 4) are very large, any comment on this? What is the influence of the snow model type used?

There is also big increase in evapotranspiration in the climate change scenario (table 4). What could explain this? And what is the influence of the relatively simple evaporation model, which is correlated to temperature and influenced by precipitation through soil moisture but does not use other input from the climate model such as changes in wind speed, cloudiness or humidity? The changes in evapotranspiration only have a limited effect the flood discharges, but the low discharges are more sensitive to these

changes.

4.4. Limitations Rcps should be added to GCMs and RCMS as source of uncertainty or limitation to the study.

---

## Referee Comment (RC2) · Anonymous Referee #2 · 6 Feb 2020

This study aims at assessing the impacts of climate change on flow patterns and patterns and frequency of floods in small ungauged rural catchments in Norway. It is an interesting manuscript that is mainly well prepared and structured. The study reveals that higher mean annual discharges are expected at the end of the 21st century. Mean annual floods are projected to increase compared to the baseline period. The manuscript's results contribute to the knowledge of climate change impacts in terms of river discharges in western Norway.

I have basically three concerns:

1. Only one RCM has been used that was driven by bias-corrected input of only

one GCM. Nowadays, a state-of-the-art climate impact study should not base its results solely on one climate simulation, but on an ensemble of climate model simulations. Moreover, the results are only discussed for RCP8.5 at the end of the 21st century. In the discussion section, the authors address this issue and put their results into a larger context (Norway study), which is important, but not done in sufficient detail. Plotting precipitation and temperature projections of the NorESM1-M / WRF model in the context of the CMIP5 or CORDEX ensemble would be helpful here. However, I am not sure whether putting the results into the context of another study justifies the approach of using only one climate simulation as input.

2. The description of the hydrological model and the regionalisation approach in the Method chapter is quite long (6 pages). I think, this is not appropriate for a manuscript claiming to investigate climate impacts. In between I wondered whether the main focus of the manuscript is actually on climate impacts or on the methodology to be applied in ungauged catchments. What I would expect to read instead is something like this: "The DDD model is a lumped, conceptual hydrological model with a module simulating the slow flow component (subsurface) and the quick component (runoff dynamics)." Additionally, something about the temporal and spatial scales, it can be applied to, that it is not fully or semi-distributed...

3. Especially in the discussion and conclusion sections, the authors mention many times the novel bias correction method that has been applied. From my point of view, instead of providing a lengthy description of the hydrological model, a description of the novel bias correction method would be much more valuable in the context of a climate impact study.

**Besides these three main concerns I have made a lot of comments on the attached RC2-supplement-pdf file.**
**General comments**

- The introduction is a bit lengthy, particularly the section between rows 70-87.

- English is usually adequate but some sentences are incomprehensible or poorly expressed, e.g. following sentence: "*The regional impacts of climate change (e.g. on local flooding) come out with the necessity of orienting adaptation measures to local climatic, geographic, economic and social conditions.*"

- The authors should consider having the manuscript revised by a native speaker.

- The authors use the term "hydroclimatic elements" meaning variables, such as precipitation and discharge. I recommend to call these "hydro-climatic variables" not elements.

**Technical**

- Multiple citations should be ordered by year.

- Equations. Many variables in the equations are not explained or mentioned in the text, some examples: Eq.2: Q(t); Eq.5: Cea; Eq.6: M

Following articles might be worth citing in some contexts in the manuscript.

- Blöschl, G. et al. (2019). Changing climate both increases and decreases European river floods. Nature

- Blöschl; G. et al (2017). Changing climate shifts timing of European floods. Science, 357, 588-590

Please also note the supplement to this comment:
https://www.nat-hazards-earth-syst-sci-discuss.net/nhess-2019-359/nhess-2019-359-RC2-supplement.pdf

————————————————————

[Figure]

**Supplement:**

[revised manuscript text omitted]

**2.3.5 Regionalizing the parameters of DDD model**

- To estimate the regionalized parameters for this study (3 hourly time step), we have used the combined method of regionalization which has been recommended for estimating regionalized DDD model parameters with hourly resolution (Tsegaw et al., 2019a). In the combined method of regionalization, we have estimated the recession parameters and critical flux using multiple regression between model parameters and CDs, and the other parameters (all in appendix 2) using
- the physical similarity method with pooled donor catchments. The parameters of the model needing regionalization are shown in appendix 1 and 2 (the bottom 5 parameters in appendix 1 and all in appendix 2).

---

## Author Response (AR1)

From:

Aynalem T. Tsegaw (Tel.: [+47] 47173764)

Knut Alfredsen, Tone M. Muthanna and Erle Kristvik

Department of Civil and Environmental Engineering, Norwegian University of Science and Technology (NTNU), S.P. Andersensvei 5, N-7491, Trondheim, Norway. E-mail addresses: aynalemtassachew1982@gmail.com, knut.alfredsen@ntnu.no, erle.kristvik@ntnu.no

tone.muthanna@ntnu.no

**and**

Marie Pontoppidan

NORCE Norwegian Research Centre, Bjerknes Centre for Climate Research, Bergen, Norway. Email address: mapo@norceresearch.no

**Subject: Cover letter accompanying resubmission of Research article (MS No.: nhess-2019-359)**

Dear Editor,

We hereby submit a revised manuscript for review for Natural Hazards and Earth System Sciences (NHESS). The paper was previously reviewed (nhess-2019-359) with a decision of major revision. We have now revised the paper in accordance with the comments from the reviewers. The details of the changes are outlined in the responses to the reviewers included with the submission.

The main comments of the first reviewer are: the results, which use input from one climate model, should be given more context in regard to other climate scenarios and some key uncertainties should be better presented, and the climate change input data should be described in more detail. *To address the comments, we have compared in detail our study results with RCP8.5 (worst case scenarios) results of other studies, and other key uncertainties are also included in the discussion section of the revised manuscript. We have also described in detail the climate change input data in the revised manuscript (section 2.2).*

The main comments of the second reviewer are: To discuss and justify the use of only one RCM that was driven by bias-corrected input of only one GCM, to reduce the lengthy description of the hydrological model and the regionalization approach and to expand the description of the novel bias correction method in the method section of the paper. *To address the comments, we have discussed in detail the comparison of the single RCM-GCM results of our study with other studies in the discussion section of the revised manuscript, and we have justified the reasons for the use of a single RCM-GCM in the responses to the reviewers. We have also revised and shortened significantly (from 6 to 2.5 pages) the hydrological model and regionalization approach and expanded the novel bias correction method (from 0.5 to 2.5pages) in the method section of the revised manuscript, section 2.2 respectively.*

We hope the major revision done in the paper could be suitable for NHESS.

Best Regards,

Aynalem T.Tsegaw

Marie Pontoppidan

Knut Alfredsen

Tone M. Muthanna

Erle Kristvik

**Hydrological impacts of climate change on small ungauged catchments-results from a GCM-RCM-hydrological model chain**

We would like to thank the reviewers for their thoughtful comments and efforts towards improving our manuscript, which have helped us to improve the quality of the manuscript. In the following, we give responses to the comments/concerns the reviewers raised.

**Reviewer 1**

**General comment**

In the manuscript the regionalized DDD model with dynamic river network was used to study climate change impacts on hydrology by 2070-2100 in ungauged catchments in the Bergen area in western Norway. Six ungauged small rural catchments are modeled with a single high-resolution downscaled climate scenario.

The manuscript is well structured and well written and bring new knowledge on estimation of climate change impacts on small ungauged basins with sub-daily time steps. However, to improve the manuscript, the results, which use input from one climate model, should be given more context in regard to other climate scenarios and some key uncertainties should be better presented.

**General Answer**

Thank you. To improve the manuscript, we have compared in detail our study results with RCP8.5 (worst case scenarios) results of other studies, and other key uncertainties are also included in the discussion section of the revised manuscript. We have also compared results of RCP8.5 and 4.5 of other studies.

**Detailed comments**

**Comment on Abstract:**

The abstract should clearly state the climate scenario (especially rcp) for which the results are based since this affects the likelihood of the proposed changes (ie. these are apparently with the rcp8.5 and are therefore likely to be the upper end of the proposed changes). The uncertainties due to use of only one climate scenario should be shortly acknowledged also in the abstract and the percentage changes of the results should be provided with less accuracy (ie. not well 256.3 % but with e.g. 260 %).

**Answer**

We have revised the abstract section of the manuscript accordingly. We have included the uncertainties due to use of only one climate scenario. The results are also provided with rounding to the nearest integer. The revision is found in the abstract, result and discussion sections of the revised manuscript. Tables, 5,7 and 9 containing decimals, are also rounded.

**Comment on Data and methods:**

More background information from the Bergen area floods and the mechanism (snow or rain or both) could be provided. In discussions the results could be reflected with this.

**Answer**

More background information on Bergen climatic conditions, floods, shifts in floods, and floods generation mechanisms are included in the revised version of the manuscript. The floods in the southern west part of Norway are mainly caused by rain in the autumn season.

**Revised Manuscript**

The included information is found in the data and methods section of the revised manuscript, lines (122 to 146).

**Comment on Section 2.2:**

Section 2.2 includes information of the climate change input data used. However, the climate change input data should be described in more detail. The climate scenario used should be more clearly stated (Global climate model, regional climate model, rcp) (NORESM-M, WSF, but the rcp is not mentioned here, apparently rcp8.5) – the corrections made to the data should be provided with more details. Is only the GCM corrected and for which variables? Is there any bias correction on the RCM data? How well do the temperatures and precipitation compare to observations? -the use of only one scenario just be better justified (since the common approach these days in to use several scenarios to enable uncertainties to be included). Why this particular model and why only one?

**Answer**

We have revised and rewritten section 2.2. The RCP used is 8.5, and the global climate model NorESM1-M (r1i1p1) output used as forcing data at the boundaries of WRF was bias corrected before the regional downscaling. We followed the approach of Bruyère et al. (2015) and corrected the monthly mean values towards the monthly mean of the reanalysis ERA-Interim (Dee et al. 2011). The correction was performed for the skin temperature and the three-dimensional pressure, humidity, temperature and the wind components. The bias correction method used is explained in detail in the data and methods section of the revised manuscript.

We agree that it is quite common to use ensembles of scenarios in climate impact studies; However, in the hydrological impact study of small catchments, the short response time of small catchments requires high temporal and spatial resolution (short duration and small special scale) climate data and getting such ensembles data is a challenge. To our knowledge, no other convective permitting, century-long, dynamically downscaled climate projections is available for Norway for short duration rainfall with a small spatial scale so that a standard ensemble analysis is not possible.

**Revised Manuscript**

The included information is found in section 2.2 of the revised manuscript, lines (160 to 227).

**Comment on Discussion:**

Since only one climate scenario is used, the influence of this decision on the results should be discussed. Table 4 states the temperature and precipitation changes with the used climate scenario (the GCM-RCM and rcp of the scenario should be added to the table header) and on page 2 the range for rcp8.5 is stated. However, these should be compared more clearly in the discussion or elsewhere (how does the chosen scenario compare to others, is it e.g. wetter than average). Also, the range of temperature and precipitation with other rcps than just 8.5 should be provided for context. The results are currently only been compared to the rcp8.5 results of e.g. NCCS report, also some comparison with the rcp4.5 could be provided for more context. How does the use of only one scenario influence the results and what are the likely results with other scenarios (e.g. are these likely to be the top end of changes in floods which can be used as worst case scenario). Currently the results, which are stated with high accuracy, can provide false sense of certainty while this major uncertainty is not well established. (The emissions used in RCP8.5 pathway are nowadays considered by some scientists as rather unlikely due to the ongoing mitigation efforts and the sinking prices of renewables. Therefore, there has been arguments against the use of this scenario as a "business as usual" scenario).

The results showed large increase in flood risk due to climate change. Other studies are referred to but the main differences between these studies explaining the differences in the results (the inclusion of different types of catchments with more snow dominated flooding and the use of several different climate scenarios) should be analyzed. The changes in max SWE (table 4) are very large, any comment on this? What is the influence of the snow model type used?

There is also big increase in evapotranspiration in the climate change scenario (table 4). What could explain this? And what is the influence of the relatively simple evaporation model, which is correlated to temperature and influenced by precipitation through soil moisture but does not use other input from the climate model such as changes in wind speed, cloudiness or humidity? The changes in evapotranspiration only have a limited effect the flood discharges, but the low discharges are more sensitive to these changes.

**Answer**

The header of table 4 has been revised accordingly i.e., the GCM, RCM and RCP are added. We have compared changes in precipitation, temperature and floods between our findings and the Norwegian Centre for Climate Services (NCCS) under Climate in Norway 2100 report (Hanssen-Bauer et al., 2015) with RCP8.5. WE have also compared the findings of the report with RCP8.5 and 4.5. The comparison has been done in the discussions section of the revised manuscript.

The comparisons of the precipitation and temperature changes with our climate model and the NCCS report with RCP8.5 show that our findings are colder and wetter than the NCCS report. The precipitation and temperature changes for Norway are 17% and 3.7°C for RCP8.5 at the end of the century (2071-2100) compared to the reference period (1971-2000). Our findings show that the mean annual changes are 22% and 3.3°C at the six study catchments in Bergen area at the end of the century compared to the reference period (1981-2011). The results are generally comparable with small differences.

**Revised Manuscript**

The included comparison is found in the discussion section (4.2.1) of the revised manuscript lines (456-464).

Detail comparisons of 200 years flood changes, between the Lawrence (2016) findings with RCP 8.5 and our findings, have been done in the revised version of the manuscript. The RCP8.5 results of the Lawrence (2016) report show slightly lower flood changes than our findings. Lawrence (2016) found that the increase in the 200 years flood is between 20% to 40% for seven of the ten study catchments used in the Hordaland county at the end of the century. Our study catchments are also located in the Hordaland county. The results of our study show that the 200-year flood changes range from 20% to 43% for five of the six the study catchments. The comparison shows that our findings are similar to the RCP8.5 report and likely at the top end of changes in floods which can be used as a worst-case scenario at the Bergen area. The slight differences in the findings are related to the temporal resolutions used (3hourly in our study and daily in the report), the bias correction methods used, the number of GCMs-RCMs used and the differences in the sizes of the catchments (less than 10km2 in our study, and 6km2 to 15499km2 in the report).

**Revised Manuscript**

The included comparison is found in the discussion section (4.3.2) of the revised manuscript, lines (620-634).

We have explained the differences in our findings and Lawrence (2016) report related to changes in flood frequencies. The main differences are the number and types of climate models, RCPs, the bias correction method, the catchment sizes and the temporal resolution used in the study. However, the comparison location is the same. We have compared our findings with results from 10 catchments in the Hordaland county (where our study catchments are located) in the report. The included comparison is found in the discussion section (4.3.2) of the revised manuscript, lines (636 - 648).

The main reason for the very large increase of changes in maximum SWE is due to a real effect of warming temperature that shifts the rain/snow boundary higher up in elevation. The other reason could be the limitations and uncertainty in estimating the snow model parameters. The snow model parameters in Distance Distribution Dynamics hydrological model are estimated from the nearby catchments which had been estimated from daily observed precipitation and temperature data in the Thomas Skaugen 2015 paper.

The big changes in actual evapotranspiration tells us that in the future period there will be more water available (to evaporate) and higher temperature (to cause evaporation) than the reference period. Yes, we agree with the reviewer in that there is a limitation with the simple evaporation model since actual evaporation is not only affected by temperature but by additional climatological factors like wind speed, humidity, cloudiness etc. This limitation could also be the other reason for a big change of actual evapotranspiration between the reference (1981-2000) and future periods (1970-2100).

**Revised Manuscript**

The limitation related to the simple evaporation model is included in the revised manuscript, lines (656-664).

4.4. Limitations RCPs should be added to GCMs and RCMS as source of uncertainty or limitation to the study.

**Answer**

We have included the limitations in the use of a single Representative Concentration Pathway (RCP8.5) in the revised manuscript.

**Revised Manuscript**

The included limitation is found in the limitation section (4.4) of the revised manuscript, lines (668-674).

**Reviewer 2**

This study aims at assessing the impacts of climate change on flow patterns and patterns and frequency of floods in small ungauged rural catchments in Norway. It is an interesting manuscript that is mainly well prepared and structured. The study reveals that higher mean annual discharges are expected at the end of the 21st century. Mean annual floods are projected to increase compared to the baseline period. The manuscript's results contribute to the knowledge of climate change impacts in terms of river discharges in western Norway.

**Three Main comments/concerns:**

1. Only one RCM has been used that was driven by bias-corrected input of only one GCM. Nowadays, a state-of-the-art climate impact study should not base its results solely on one climate simulation, but on an ensemble of climate model simulations. Moreover, the results are only discussed for RCP8.5 at the end of the 21st century. In the discussion section, the authors address this issue and put their results into a larger context (Norway study), which is important, but not done in sufficient detail. Plotting precipitation and temperature projections of the NorESM1-M / WRF model in the context of the CMIP5 or CORDEX ensemble would be helpful here. However, I am not sure whether putting the results into the context of another study justifies the approach of using only one climate simulation as input.

**Answer**

Since the new climate simulation is specially made for southwest coast of Norway with a time resolution of 3 hours and 4km by 4km spatial resolution, the comparison with other studies is interesting to see how the new method performs compared to what we know before and if it seems reasonable. Therefore, we believe that putting our results into the context of another study justifies the approach of using only one climate simulation as input.

We agree that the state-of-the-art of climate impact study normally is based on an ensemble of climate model simulations; however, for climate impact studies on small catchments, we have used only one GCM-RCM model simulations. We have used a single GCM-RCM model simulation is we actually can not find sub-daily data which is required for small catchments (< 10km2) study. We know that improvement is needed for the west coast of Norway so that we provide something brand new sub-daily climate data and bias corrected GCM data as input. The reasons for using a single GCM-RCM model simulation are summarized as follows:

- i) The single GCM-RCM simulation used in this study was generated based on the need to improve the precipitation distribution for the west coast of Norway, this was done through the new bias correction method and by utilizing kilometre-scale resolution.
- *ii)* Limited computational resources to conduct ensembles of downscaling. This 60-year GCM-RCM downscaling alone has used approximately 650,000 cpu hours on a national hpc machine.

We have discussed in detail the results obtained using a single scenario and single climate model into a larger context in the discussion section of the revised manuscript. The discussions are found on lines 445 to 483 and lines 620 to 648 in the discussion section.

2. The description of the hydrological model and the regionalization approach in the Method chapter is quite long (6 pages). I think, this is not appropriate for a manuscript claiming to

investigate climate impacts. In between I wondered whether the main focus of the manuscript is actually on climate impacts or on the methodology to be applied in ungauged catchments. What I would expect to read instead is something like this: "The DDD model is a lumped, conceptual hydrological model with a module simulating the slow flow component (subsurface) and the quick component (runoff dynamics)." Additionally, something about the temporal and spatial scales, it can be applied to, that it is not fully, or semi distributed...

**Answer**

Thank you. We have revised and shortened significantly (from 6 to 2.5 pages) the hydrological model and regionalization approach in the method section of the manuscript (section 2.3). We have included only the main points about DDD model and the regionalization methods.

**Revised Manuscript**

The revision is found in section 2.3 of the revised manuscript, lines (241 to 294).

3. Especially in the discussion and conclusion sections, the authors mention many times the novel bias correction method that has been applied. From my point of view, instead of providing a lengthy description of the hydrological model, a description of the novel bias correction method would be much more valuable in the context of a climate impact study.

**Answer**

Thank you. We have shortened the lengthy description of the hydrological model (from 6 pages to 2.5 pages) and expanded the novel bias correction method (from 0.5 page to 2.5 pages) under section 2.3 and 2.2 respectively.

**Revised Manuscript**

The revisions are found in section 2.3 and section 2.2 respectively in the revised manuscript.

Besides these three main concerns I have made a lot of comments on the attached RC2-supplement-pdf file.

**General comments**

1. The introduction is a bit lengthy, particularly the section between rows 70-87.

**Answer**

The introduction is shortened. Particularly, section 70-87 is reduced significantly (from 70-87 to 71-82).

2. English is usually adequate, but some sentences are incomprehensible or poorly expressed, e.g. following sentence: "The regional impacts of climate change (e.g. on local flooding) come out with the necessity of orienting adaptation measures to local climatic, geographic, economic and social conditions."

**Answer**

We went through the manuscript and corrected the poorly expressed sentences.

3. The authors should consider having the manuscript revised by a native speaker.

**Answer**

We revised the language, and we believe that the language is enough for publication.

4. The authors use the term "hydroclimatic elements" meaning variables, such as precipitation and discharge. I recommend calling these "hydro-climatic variables" not elements.

**Answer**

Thank you. We have changed hydroclimatic elements to hydro-climatic variables in the revised manuscript.

**Technical**

1. Multiple citations should be ordered by year.

**Answer**

Thank you, we have revised accordingly.

 Equations. Many variables in the equations are not explained or mentioned in the text, some examples: Eq.2: Q(t); Eq.5: Cea; Eq.6: M

Answer

We have explained the variables of the equations in the revised manuscript.

Following articles might be worth citing in some contexts in the manuscript.

• Blöschl, G. et al. (2019). Changing climate both increases and decreases European river floods. Nature

• Blöschl; G. et al (2017). Changing climate shifts timing of European floods. Science, 357, 588-590 C3

**Answer**

*Thank you. We have included the references in the introduction (lines 45-47), study area (lines 143-146) and discussion (lines 554-556) sections of the revised manuscript.*

[revised manuscript text omitted]

---

## Editor Decision (ED1)

**Editor comments to: "Hydrological impacts of climate change on small ungauged catchments – results from a GCM-RCM-hydrologic model chain"**

June 5, 2020

**1 Introduction**

- p3 l58: Adverse effects ... this sentence is still strange as "effects" is plural and the related verb "calls" is singular. Furthermore, the "," doesn't seem to be right.

- p3 l61: "The projected increase ..." is still not a correct sentence. The subject "projected increase" uses the verb "increases". That sounds strange. Suggestion: "An increase in heavy localized precipitation events as projected suggests an increase in precipitation generated local flooding"

- p5 l110ff: I have the same question as the reviewer. Even if you add a reference, it seems odd that you quantify model performance for an ungauged catchment as by definition of "ungauged", there is no data to quantify the performance. I think this needs to be stated somewhat differently.

**2 Data and Methods**

The description of the bias correction method on page 12/13 is still strange. I give you an example for a way of writing this, starting at your l.213.

**2.1 Example for anomaly based bias adjustment in your section 2.2.1**

A reference period of 30-years (ref, from 1981 to 2010) was selected for which we calculate mean values for the month $m = 1, \ldots, 12$ of a variable $Y_{\text{ref}}^{\text{ERA}}$ (e.g. temperature) from ERA-Interim

$$\overline{Y_{\text{ref},m}^{\text{ERA}}} = \frac{1}{N_m} \sum_{i=1}^{N_m} Y_{\text{ref},i}^{\text{ERA}} \; . \tag{1}$$

Analogously, we calculate monthly means for variables $Y_{\text{ref}}^{\text{Nor}}$ from the NorESM-1-M simulations, denoted as $\overline{Y_{\text{ref},m}^{\text{Nor}}}$.

For NorESM-I simulations $Y_{\text{ref},i}^{\text{Nor}}$ (six hourly time resolution) in the reference period, we define anomalies for a data point $i$ as

$$Y_{\text{ref},i}^{\prime,\text{Nor}} = Y_{\text{ref},i}^{\text{Nor}} - \overline{Y_{\text{ref},m}^{\text{Nor}}} \tag{2}$$

for each month $m$ as deviations from the monthly means of the reference period. For simulations $Y_{\text{fut},i}^{\text{Nor}}$ for the future period, anomalies are also taken as deviations from monthly means of the reference period

$$Y_{\text{fut},i}^{\prime,\text{Nor}} = Y_{\text{fut},i}^{\text{Nor}} - \overline{Y_{\text{ref},m}^{\text{Nor}}} \,. \tag{3}$$

An anomaly based bias adjustment for a data point $i$ in month $m$ is now defined as

$$\widehat{Y}_{\text{per},i}^{\text{Nor}} = Y_{\text{per},i}^{\text{Nor}} - \overline{Y_{\text{ref},m}^{\text{Nor}}} + \overline{Y_{\text{ref},m}^{\text{ERA}}} \,, \tag{4}$$

where the index per denotes the reference (ref) or future (fut) period.

**3  Miscellaneous**

- write $\sum_{i=1}^{N}$ instead of $\sum_{i=1}^{i=N}$

- Until the late 80ties, the "*" was used on typewriters as a symbol for multiplication as there was no dedicated symbol on the typerwriters. With modern computer typesetting systems, this is not needed anymore. For denoting $a$ times $b$ either use the convention that no sign means multiplication ($a\,b$) or, if needed, write $a \cdot b$. Your Eq. 6, using the above notation, should look like

$$\Delta Y = \frac{Y_{\text{fut}} - Y_{\text{ref}}}{Y_{\text{ref}}} \, 100 \,. \tag{5}$$

---

## Author Response (AR2)

From:

Aynalem T. Tsegaw (Tel.: [+47] 47173764)

Knut Alfredsen, Tone M. Muthanna and Erle Kristvik

Department of Civil and Environmental Engineering, Norwegian University of Science and Technology (NTNU), S.P. Andersensvei 5, N-7491, Trondheim, Norway. E-mail addresses: aynalemtassachew1982@gmail.com, knut.alfredsen@ntnu.no, erle.kristvik@ntnu.no tone.muthanna@ntnu.no

and

Marie Pontoppidan

NORCE Norwegian Research Centre, Bjerknes Centre for Climate Research, Bergen, Norway. E-mail address: mapo@norceresearch.no

**Subject: Cover letter accompanying resubmission of Research article (MS No.: nhess-2019-359)**

Dear Editor,

We hereby submit a revised manuscript for review for Natural Hazards and Earth System Sciences (NHESS). The paper was previously reviewed (nhess-2019-359) with a decision of minor revision. We have now revised the paper in accordance with the comments from the second reviewer. The details of the changes are outlined in the responses to the reviewer included with the submission.

The main comments/concerns of the second reviewer are: the difficulties to understand the bias correction method in the methodology section of the manuscript, comments made in the first PDF were not addressed satisfactorily, and language and presentation of results are not precise.

To address the comment related to bias correction method, we have revised the whole paragraph and the mathematical equations in the revised manuscript. We have explained how the monthly mean time series was computed and how the deviations were calculated in the revised paragraph.

To address the comments related to comments made in the first PDF, we have re-considered and addressed comments made in the first PDF (listed here and not listed here by the reviewer) in the revised manuscript.

To address the comment related to language and presentation of results, we went through the whole manuscript carefully and revised sentences which had wrong grammatical expressions.

We hope the revisions done in the paper could be suitable for NHESS.

Best Regards,

Aynalem T.Tsegaw

Marie Pontoppidan

Knut Alfredsen

Tone M. Muthanna

Erle Kristvik

**Hydrological impacts of climate change on small ungauged catchments-results from a GCM-RCM-hydrological model chain**

We would like to thank reviewer # 2 for the second-round comments and efforts towards improving our manuscript, which have helped us to improve the quality of the manuscript. In the following, we give responses to the comments/concerns the reviewer raised.

**Reviewer # 2**

**General comment**

The manuscript has improved with the first revision, especially with regard to a better embedding of the results with other research on climate change impacts in Norway and also the shortening of the DDD-model description is appreciated. However, further improvements are required. What I find somewhat discouraging here from the point of view of the reviewer is that many comments I made in the first PDF were simply ignored. I will only list a few examples here and expect that if the authors get the chance to resubmit the manuscript, the comments of the first round will also be taken into account.

*General Answer*

*Thank you. We have re-considered and addressed comments made in the first PDF. We have also revised the manuscript based on the new comments, which helped us for improving the manuscript.*

**Detailed comments**

1) **Examples of comments from the first round of reviews not addressed by authors**

On page 3, line 57. It was already commented in the previous review that this sentence is not really elegant: "Climate change adverse results upon streamflow regimes worldwide..." Better start like this: "Adverse effects of climate change on river regimes worldwide..."

There is still the sentence in the introduction (page 3, line 61): "Projected increase in the frequency and intensity of heavy localized precipitation events, based on climate models, contributes to increasing in precipitation-generated local flooding, ..."To me, this sounds like the "projected increase of heavy localized precipitation events" itself cause an increase of local flooding in reality. Maybe it is simply the term "contributes", which is not used properly in this context.

Page 4, line 84: "Climate impact assessment on hydrology ... is challenging..." More correct or precise: "Assessing the impacts of climate change on hydrology...is callenging..." Moreover, the sentence is way too long.

*Answer*
*We have revised the sentence on page 3, line 57 to make the sentence elegant as suggested by the reviewer. The revised sentence is found on page 3, lines 58-59 in the revised manuscript.*

*We have revised the sentence on page 3, line 61 to make the sentence precise as suggested by the reviewer. The word contribute has been removed. The revised sentence is found on page 3, lines 62- 64 in the revised manuscript.*

*We have revised the sentence on page 4, line 84 to make the sentence more correct. We kept the long sentence as it is because it contains important information on the challenges of climate impact study on hydrology of small ungauged catchments. The revised sentence is found on page 4, lines 83-87 in the revised manuscript.*

*We have also re-considered and addressed other comments made in the first PDF (not listed here by the reviewer) in the revised manuscript. The revisions are highlighted with yellow in the revised manuscript.*

**2) Description of bias correction method**

page 9 (bottom) and page 10 (top): From the description, it is not clear to me how the bias correction was performed. I could only guess.

The sentence: "For the reference period the 6-hourly NorESM1-M input data were split into a monthly mean term and the deviation from this." is not clear.

First of all, the reader does not know whether the mean monthly time series is composed of twelve values, representing the 30-years mean or if it is a time series composed of 12*30 values. I would also not use the term "split into". Explain first how the monthly mean time series was computed and then how the deviations were calculated. The equations 1-3 are not referred to in the text and are not self-explanatory, because the variabels are not explained. Which variable represents the mean, which the deviation?

*Answer*

*Thank you. To address the concern related to the difficulties in understanding of the bias correction method specifically the mathematical description part on pages 8 and 9, we have revised the whole paragraph and the mathematical equations in the revised manuscript. We have explained how the monthly mean time series was computed and how the deviations were calculated in the revised paragraph. We have revised and increased the equations from 3 to 4 and described them in the body of the manuscript. The revision is found on page 12(bottom) and 13(top), lines 212-235 in the methodology section of the revised manuscript.*

**3) Proper and / or imprecise use of English language**

It seems to me that the manuscript was not corrected by a native speaker. There are still sentences, which are grammatically not entirely correct. Sometimes it is only a missing "the" ...

*Answer*

*To address the concern related to language, we went through the whole manuscript carefully and revised sentences which had wrong grammatical expressions.*

**4) Discussion**

It seems that there is quite some redundancy between Discussion and Results and sometimes one and the same results are referred to too often. For example: "the magnitude of the 200-year flood changes range from 16% to 43%" (page 28, line 610 and page 29, line 631). So, there is some space for improvement.

*Answer*

*Thank you. We have removed the sentence, which is found on page 28, line 610. The sentence on page 29, line 631 has been moved to the top and revised. The revision is found on page 38, lines 655-660 in the revised manuscript and highlighted with yellow. The repetitions of results in the discussions have been removed in the revised manuscript.*

**Specific comments**

**1)** Page 2, line 26: RCPs are not "emission" scenarios but"greenhouse gas concentration" scenarios. It is also used somewhere else in the text.

*Answer*

*Thank you. We have removed the word "emission" from whole manuscript in the revised version. Therefore, it will be read as RCP8.5 or 4.5 scenario.*

**2)** page 5, line 110: How can the performance of a simulation been measured if there are no observations? "Even if the DDD model predicts flow at ungauged catchments satisfactory 0.5<=Kling-Gupta Efficiency < 0.75)..."

*Answer*

*We have already evaluated the performance of DDD model in predicting flows at small ungauged catchments in Norway using observed flow data with a recently published paper (Tsegaw et al.,2015). From the model test results, we know that the model predicts flow at ungauged small catchments satisfactorily in Norway. We have included the reference in the revised manuscript. The reference is found on page 5, line 113 in the revised manuscript.*

**3)** Page 6, line 137 "...dominated by rain floods..." I think, "rain flood" is not a proper term, it is either a river flood, if you want to distinguish from e.g. "coastal flood" or a flood caused by (heavy) rainfall.

*Answer*

*The sentence with the word "...dominated by rainfall" has been revised and replaced by "...a rainfall-dominated floods ....". The revision is found on page 6, line 138 in the revised manuscript.*

**4)** page 9, line 207: Reference Bruyere et al. 2015 is not in the references. Is it Bruyere et al. 2014?

*Answer*

*Thank you. We have put the correct reference and removed Bruyere et al. 2014. The correct reference is Bruyère, C. L., Monaghan, A. J., Steinhoff, D. F., and Yates, D. Bias-Corrected CMIP5 CESM Data in WRF / MPAS Intermediate File Format. NCAR Technical Note NCAR/TN-515+STR, 2015. The included reference is found in the reference section of the revised manuscript.*

**5)** page 11, line 246: Be more specific: "catchments from small to large", what does it mean in terms of km2? "Temporal resolution from low to high", what is low what is high?

*Answer*

*We have described in detail by giving examples to small, large, high and low terms in the revised manuscript. The revision is found on page 14, lines 257-258 in the revised manuscript.*

**6)** page 14: The brackets in equation 5 are in my opinion superfluous.

*Answer*

*We agree that with and without the brackets have the same meaning, however; we kept the bracket as it is to make the equation clearer.*

**7)** page 14, line 315-320: You could simply say that the 3-hourly time series were converted to annual time series instead of writing 5 lines.

*Answer*

*Thank you. We have shortened the bottom four lines of the paragraph into two lines. The revision is found on page 19, lines 344-345 in the revised manuscript.*

8) page 17, lines 388f. ...temperature rises "in" <-- "by". The mean annual flow increases "from" <-- "between".

*Answer*

*Thank you. We have revised the sentences as per the reviewer suggestion. The revisions are found on page 22, lines 416 - 417 in the revised manuscript.*

**9)** page 19, line 410: Abbreviation "CV" not introduced before usage
*Answer*

*Thank you. We wrote centre of volume in the methodology section but forgot to include CV under bracket. We have introduced CV under bracket in the revised manuscript i.e. centre of volume (CV). The revision is found on page 20, line 370 in the methodology section of the revised manuscript.*

**10)** page 22, line 471: "moderate increase in temperature". 3 degrees increase is not moderate in

my point of view, only if you compare it to other world regions, where increase in air temperature might be much higher at the end of the 21st century.

*Answer*

*Thank you. We have removed "moderate" and "substantial" words from the sentence, and the sentence has been revised. The revision is found on page 32 and 33, lines 525 -526 in the revised manuscript.*

**11)** page 24, line 515: ...increase of...-9% to 17%... is it really -9%, if yes, increase would be the wrong term

*Answer*
*The sentence has been revised. It is actually -9%, and we have rephrased it as "....will reduce up to 9%". The revision is found on page33, lines 563-565 in the revised manuscript.*

**12)** page 28, line 610: You present the value of the 50-year flood by another study and compare it to the 200-year flood from your study. Why don't you refer to the 50-year flood value from your study?

*Answer*
*Thank you. We have made minor revisions to the sentences and paragraphs under **4.3.2 i.e.***

***Changes in flood frequency.** We put the summary of our findings in the first paragraph, and we focused on the findings of other studies and the comparison with our findings in the rest paragraphs. The revisions are found on pages 37 , lines 653 in the discussion section of the revised manuscript.*

**13)** page 29, line 642: "...and perhaps this is the data..." Don't you know or why are you saying "perhaps"? I recommend to reformulate this and to be more precise.
*Answer*
*We have reformulated the sentence and perhaps is removed. The revision is found on page 39, lines 695-697 in the discussion section of the revised manuscript.*

**14)** page 30, line 659: "...will be higher water available" <-- more water

*Answer*

*We have replaced higher with more as suggested by the reviewer in the revised manuscript. The revision is found on page 40, lines 713 in the limitation section of the revised manuscript.*

**15)** page 31, line 678: "...that we are not totally off..." <-- This is colloquial language

*Answer*

Thank you. We have removed the phrase "we are not totally off" and the sentence consisting of the phrase has been revised. The revision is found on page 41, lines 732-734 in the conclusion section of the revised manuscript.

[revised manuscript text omitted]

215  January to December, resulting in 12 monthly mean terms (Eq. 1). Similarly, the 12 monthly means were calculated from the 6-hourly NorESM1-M input data for the reference period, and subsequently the 6-hourly input data was separated into a monthly mean term, and the deviation from the monthly mean term (Eq. 2). For the future period, the 6-hourly NorESM1-M input data was separated into the monthly mean from the reference period, and the deviation from this,

220  leaving the climate change signal and the non-stationary variability in the deviation term (Eq. 2). The new bias corrected ($BC$) NorESM1-M data were then calculated by substituting the NorESM monthly mean with the ERAI mean (Eq. 3), which is equivalent to the summation of the monthly mean from the reanalysis product and the NorESM1-M deviation term (Eq. 4).

$$\overline{ERAI_{ref}} = \frac{\sum_{n=1}^{n=k} ERAI_{m,n}}{k} \tag{1}$$

225  where $m$ is a month from January to December, $n$ is the number of time steps in a month (n = 1, 2, 3…, k), k is the total number of time steps in a month.

$$NorESM = \overline{NorESM_{ref}} + NorESM' \tag{2}$$

$$NorESM_{BC} = NorESM - \overline{NorESM_{ref}} + \overline{ERAI_{ref}} \tag{3}$$

$$NorESM_{BC} = \overline{ERAI_{ref}} + NorESM'  \qquad (4)$$

Here, the mean terms ($\overline{ERAI_{ref}}$ and $\overline{NorESM_{ref}}$) were 
[revised manuscript text omitted]

---

## Author Response (AR3)

From:

Aynalem T. Tsegaw (Tel.: [+47] 47173764)

Knut Alfredsen, Tone M. Muthanna and Erle Kristvik

Department of Civil and Environmental Engineering, Norwegian University of Science and Technology (NTNU), S.P. Andersensvei 5, N-7491, Trondheim, Norway. E-mail addresses: aynalemtassachew1982@gmail.com, knut.alfredsen@ntnu.no, erle.kristvik@ntnu.no tone.muthanna@ntnu.no

and

Marie Pontoppidan

NORCE Norwegian Research Centre, Bjerknes Centre for Climate Research, Bergen, Norway. E-mail address: mapo@norceresearch.no

**Subject: Cover letter accompanying resubmission of Research article (MS No.: nhess-2019-359)**

Dear Editor,

We hereby submit a revised manuscript for review for Natural Hazards and Earth System Sciences (NHESS). The paper was previously reviewed (nhess-2019-359) with a decision of minor revision. We have now revised the paper in accordance with the comments from the editor. We have included all the suggestions of the editor in the revised manuscript. The details of the changes are outlined in the responses to the editor included with the submission.

We hope the revisions done in the paper could be suitable for NHESS.

Best Regards,

Aynalem T.Tsegaw

Marie Pontoppidan

Knut Alfredsen

Tone M. Muthanna

Erle Kristvik

**Hydrological impacts of climate change on small ungauged catchments-results from a GCM-RCM-hydrological model chain**

We would like to thank the editor for comments and efforts towards improving our manuscript. In the following, we give responses to the comments/concerns the editor raised.

**Editor comments and answers**

**1) Introduction**

**1.1)** p3 l58: Adverse effects … this sentence is still strange as "effects" is plural and the related verb "calls" is singular. Furthermore, the "," doesn't seem to be right.

*Answer*
*We have revised the sentence on page 3, line 58 to make the sentence grammatically correct and the "," has been removed. The revised sentence is found on page 3, lines 58-59 in the revised manuscript.*

**1.2)** p3 l61: "The projected increase ..." is still not a correct sentence. The subject "projected increase" uses the verb "increases". That sounds strange. Suggestion: "An increase in heavy localized precipitation events as projected suggests an increase in precipitation generated local flooding".

*Answer*
*We have revised the sentence on page 3, line 61 to make the sentence correct as suggested by the editor. The revised sentence is found on page 3, lines 62-63 in the revised manuscript.*

**1.3)** p5 l110ff: I have the same question as the reviewer. Even if you add a reference,

it seems odd that you quantify model performance for an ungauged catchment as

by definition of "ungauged", there is no data to quantify the performance. I think

this needs to be stated somewhat differently.

*Answer*

*Thank you. We have re-stated the sentences differently on page 5, line 110 to make the sentences*

*correct. The revision is found on page 5 lines 109-113 in the revised manuscript.*

**2) Data and Methods**

The description of the bias correction method on page 12/13 is still strange. I give you

an example for a way of writing this, starting at your l.213.

*Answer*

*Thank you. We have revised the paragraphs on pages 12 and 13 as suggested by the editor and*

*the revision is found on pages 12 and 13, lines 206-224 in the revised manuscript.*

**3 Miscellaneous**

- write $\sum_{i=1}^{N}$ instead of $\sum_{i=1}^{i=N}$

- Until the late 80ties, the "*" was used on typewriters as a symbol for multiplication as there was no dedicated symbol on the typerwriters. With modern computer typesetting systems, this is not needed anymore. For denoting $a$ times $b$ either use the convention that no sign means multiplication $(a\,b)$ or, if needed, write $a \cdot b$. Your Eq. 6, using the above notation, should look like

$$\Delta Y = \frac{Y_{\text{fut}} - Y_{\text{ref}}}{Y_{\text{ref}}} 100 \, . \tag{5}$$

*Answer*

*Thank you. We have revised the summation, equation 6 and 7 as the editor suggested. The revisions are found on page 12 line 210; page 18, lines 324-327 (equation 6) and page 20, line 352 (equation 7).*

[revised manuscript text omitted]